# Association between hypertension and impaired lung function among adults: A systematic review and meta-analysis

Dilakshi Lekamge[1]*, Anuradhani Kasturiratne[1], Malay Kanti Mridha[2], John Chambers[3,4]

1 Department of Public Health, Faculty of Medicine, University of Kelaniya, Ragama, Sri Lanka, 2 Centre for Non-communicable Diseases and Nutrition, BRAC James P Grant School of Public Health, BRAC University, Dhaka, Bangladesh, 3 Department of Epidemiology and Biostatistics, School of Public Health, Imperial College London, London, United Kingdom, 4 Population and Global Health, Lee Kong Chian School of Medicine, Nanyang Technological University, Singapore, Singapore

* dilakshi.lekamge@gmail.com

## Abstract

### Background

Impaired lung function and hypertension are both significant global public health concerns, and their potential association has recently drawn considerable interest. This systematic review and meta-analysis aims to investigate the association between impaired lung function and hypertension, and vice versa.

### Methods

We searched CINAHL (EBSCOhost), MEDLINE (PubMed), Scopus and Web of Science up to 22nd July 2025 for observational studies involving adults (≥ 18 years) reporting hypertension and impaired lung function. Grey literature was also searched, and data extraction was verified independently. The review was registered with PROSPERO (CRD42023427631). Random effects models estimated pooled ORs with 95% CIs. Subgroup analysis and meta-regression were performed to identify potential sources of heterogeneity. Quality assessment was conducted using Newcastle-Ottawa Scale and JBI checklist and publication bias were appraised.

### Results

This systematic review included 28 articles, of which 26 articles were selected for the meta-analysis. Of these, 17 studies (n = 134823) examined impaired lung function as the exposure, while 9 studies (n = 12522) investigated hypertension as the exposure. Our findings revealed that the pooled unadjusted effect of lung function impairment on hypertension was 1.70 (95% CI: [1.53–1.88]), while the pooled adjusted effect was lower at 1.40 (95% CI: [1.31–1.49]). In contrast, the pooled unadjusted effect of hypertension on lung impairment was 3.00 (95% CI: [1.86–4.82]), but the pooled

**Data availability statement:** All relevant data are within the paper and its Supporting Information files.

**Funding:** This work was supported by the National Institute for Health and Care Research (NIHR) (grant numbers 16/136/68 to JC and 132960 to JC, AK, MKM) using UK international development funding from the UK government to support global health research. The funder had no role in study design, data collection and analysis, decision to publish, or preparation of the manuscript.

**Competing interests:** The authors have declared that no competing interests exist.

adjusted effect decreased to 1.94 (95% CI: [1.51–2.50]). The findings indicate that obstructive, restrictive and mixed lung function impairments are significantly associated with hypertension. Restrictive lung impairment consistently shows a stronger association with hypertension compared to obstructive impairment when impaired lung function is the exposure.

## Conclusion

This study concludes that there is a significant positive association between impaired lung function and hypertension in both directions, with stronger temporal evidence suggesting that impaired lung function may precede hypertension, while the reverse causal direction lacks such evidence.

---

## Introduction

Hypertension is a widely recognized leading risk factor for cardiovascular disease [1] and a growing global public health concern, being a significant cause of morbidity and mortality [2]. Hypertension typically remains asymptomatic until complications arise, as long as the tissues continue to receive adequate blood supply. As a result, the condition often goes undetected until the occurrence of a precipitous event, unless blood pressure is regularly monitored.

It is reported that global burden of high blood pressure is over 1 billion individuals [1] accounting for an estimation of 17.9 million lives annually or 31% of global deaths [3].

Globally, the prevalence of hypertension has been steadily rising. In 2000, 26.4% of adults were affected, with predictions indicating a 60% increase by 2025, reaching 29.2%, or approximately 1.56 billion people [4]. By 2008, around 40% of adults aged 25 and above worldwide had high blood pressure, including those on antihypertensive medication, with the highest prevalence in the African Region (46%) and the lowest in the American region (35%) [5]. A later estimate found that the global prevalence of hypertension among adults to be 30–45% [6,7], with age-standardized prevalence rates of 24% in men and 20% in women in 2015 [7,8]. By 2025, hypertension is estimated to cause 7.5 million deaths, resulting 57 million disability-adjusted life years (DALYs) [4].

It is imperative to consider early detection of hypertension in broader perspective to address the significant health and economic burden of cardiovascular diseases as yet preventive assessments typically begin only after the age of 40 years [9]. It has been identified that hypertension is responsible for an estimated 54% of all strokes and 47% of all ischemic heart disease globally [10] and with adequate treatment the risk of major cardiovascular events including stroke, coronary artery disease, and congestive heart failure [11] can be reduced.

Pulmonary function tests are essential in clinical practice for identifying the functional status of the lungs including airway obstruction and other ventilatory defects [12], with spirometry being the most commonly performed standard test to assess

lung function by measuring the rate and volume of expired air [13], providing the key outcomes such as forced expiratory volume in one second (FEV$_1$), forced vital capacity (FVC) and the FEV$_1$/FVC [12]. Lung function measured by these spirometric indices depends on height, weight, age and gender [14] acting as an important predictor of morbidity and mortality [15], declining slowly throughout the adult life, even in healthy individuals with an accelerated decline often observed after 70 years [14].

Obstructive airway diseases (OADs) include asthma and chronic obstructive pulmonary disease (COPD) [16], whereas alteration in the parenchyma, pleura, and chest wall [17] result in restrictive lung function impairment causing several underlying conditions and diseases, such as interstitial lung diseases, pleural effusions, neuromuscular diseases, thoracic deformities and obesity [18], which can be expressed as a reduction in FVC with normal FEV$_1$/FVC.

Being a major cause of global morbidity and mortality, Chronic obstructive pulmonary disease (COPD) is considered to be the third leading cause of death worldwide in 2030 [19]. Even though smoking is the major cause of COPD, occupational exposures, biomass exposure and exposure to air pollutants have substantial effects on the development of OADs resulting lung function impairment, especially those living in Asia and Africa [16,20].

Several studies have demonstrated the interaction between the respiratory and cardiovascular systems in pathophysiological contexts since the early 1990s [21,22], with Forced Vital Capacity (FVC) serving as a significant predictor of future cardiovascular disease and hypertension, irrespective of the underlying pathophysiological mechanisms [23–26]. Chronic obstructive pulmonary disease (COPD) characterized by airflow obstruction often co-exists with cardiovascular disease (CVD) and hypertension [27]. Moreover, restrictive lung impairment has also been linked to several cardiovascular disease (CVD) risk factors, such as obesity [28], diabetes mellitus [29], dyslipidemia [30], and hypertension [31].

Hypertension and impaired lung function are both significant public health concerns, and their potential association has recently drawn considerable interest. Several epidemiological studies have revealed the association between hypertension and its impact on lung function [16,24,32–39]. Moreover, evidence also suggests that individuals with lung function impairment may have a risk of developing hypertension [40]. However, some researchers found no evidence of an association between airflow obstruction and hypertension [41]. Some studies have revealed that impaired lung function could be attributed to the drugs used to treat hypertension, particularly beta blockers, rather than hypertension itself [37]. However, other researchers have suggested a contrasting point of view, emphasizing that hypertension risk could also be modulated by the prolong use of higher doses of inhaled steroids for treating chronic respiratory conditions [42]. Nonetheless, studies have identified that age and smoking habits accounted for the association between cardiovascular disease and hypertension with airflow obstruction in the general population [27].

To date there has not been either a systematic review or a meta-analysis to explore the association between hypertension and impaired lung function and vice versa. Therefore, we used the available data from cross-sectional, prospective longitudinal, and retrospective cohort studies to examine the literature and to perform a systematic review to ascertain the association between hypertension and impaired lung function and to quantify such association using meta-analysis techniques.

## Materials and methods

Systematic review of the literature and meta-analysis were conducted in accordance with the PRISMA (Preferred Reporting Items for Systematic reviews and Meta-Analyses) guidelines [43]. The protocol for this systematic review and meta-analysis was pre-registered on the international prospective register of systematic reviews (PROSPERO 2023 CRD42023427631) and is available at: https://www.crd.york.ac.uk/PROSPERO/view/CRD42023427631 (S1 File).

### Search strategy

We searched for relevant studies published from the database inception to 22$^{nd}$ July 2025 from four bibliographic databases namely CINAHL (EBSCOhost), MEDLINE (PubMed), Scopus, and Web of Science. The searches were conducted

without any language restrictions and limited to human research. Unpublished studies were also sought. Grey literature was searched through Google Scholar, ProQuest, and snowballing method, where the references of relevant papers were hand-searched. The searches were rerun prior to the final analysis, and any further studies were identified and retrieved for inclusion.

The following search strategy was developed in order to conduct the systematic literature search and capture relevant records from each of the four databases: (Hypertension OR "high blood pressure" OR "elevated blood pressure" OR "increased blood pressure" OR "raised blood pressure" OR "Hypertens*") AND ("forced expiratory volume" OR "forced vital capacity" OR "timed vital capacity" OR "vital capacity" OR "spirometry" OR "respiratory physiological phenomena*" OR "respiratory physiological phenomenon*" OR "lung pathophysiology" OR "total lung capacit*" OR "impaired lung function" OR "declined lung function" OR "reduced lung function" OR "limited lung function" OR "reduced spirometry" OR "impaired spirometry" OR "reduced FEV1" OR "reduced forced expiratory volume in 1 second" OR "reduced FVC" OR "reduced forced expiratory volume" OR "reduced VC" OR "reduced forced vital capacity" OR "FEV1/FVC ratio") AND ("adult*" OR "elder*"). A simplified search strategy was used to search the relevant other sources: (hypertension AND "impaired lung function").

### Selection criteria

Studies that met the following criteria were included in this systematic review and meta-analysis: (1) observational studies (cohort, case-control and cross-sectional studies); (2) studies that included adult males and/ or females ≥18 years who have been recruited in the study irrespective of their comorbidities; (3) studies that identified the incidence or prevalence of hypertension in adults with impaired lung function or impaired lung function in adults with hypertension; (4) studies that found an association between hypertension and impaired lung function or vice versa and (5) studies based on spirometric measurements to assess lung function.

We excluded (1) articles of reviews (literature reviews, systematic reviews, and meta-analyses), qualitative studies, trials registers, case reports, case series, book chapters, letters to editors, protocols, and conference proceedings; (2) studies comprising patients with acute respiratory diseases (Example: Bronchitis) and physical disability that may affect lung function (scoliosis and kyphosis); (3) studies of patients contraindicated for undergoing spirometry (recent myocardial infarction, heart failure), surgery to eye (cataract surgery, glaucoma), chest or abdomen done less than 3 months prior to the study; (4) studies involving pregnant women; (5) articles in which the definition of hypertension and impaired lung function remain unclear and (6) studies that do not mention the method of measuring lung function.

The articles were selected based on the definition of hypertension as systolic blood pressure (SBP) ≥ 140 mmHg and/ or diastolic blood pressure (DBP) ≥ 90 mmHg, or the use of antihypertensive medication at the time of examination or self-reported history of hypertension or physician diagnosis of hypertension. This definition was considered irrespective of the type of device used (mercury sphygmomanometer or digital blood pressure monitor), the time of measuring the blood pressure (daytime, night-time, or over a continuous 24-hour period), the number of blood pressure readings taken (two or three blood pressure readings in succession) and the measurement location (clinic, home or office). We only accepted the articles where impaired lung function was defined according to the criteria used in the individual studies, provided that impairment was based on $FEV_1$ (forced expiratory volume in 1 second), FVC (forced vital capacity) and $FEV_1$/FVC spirometric measurements. Studies were included if they reported impaired lung function as obstructive, restrictive, or mixed pattern, or if impairment was defined using accepted thresholds such as $FEV_1$/FVC < 0.70 or values below the lower limit of normal (LLN, < 5th percentile) of the $FEV_1$/FVC ratio calculated from the reference equations. No interventions were considered.

### Screening

All records identified during the database search were pooled and the duplicates were removed using the Zotero 7.0.11 reference management software. The PRISMA flow diagram was used to summarize the selection process (Fig 1). Titles

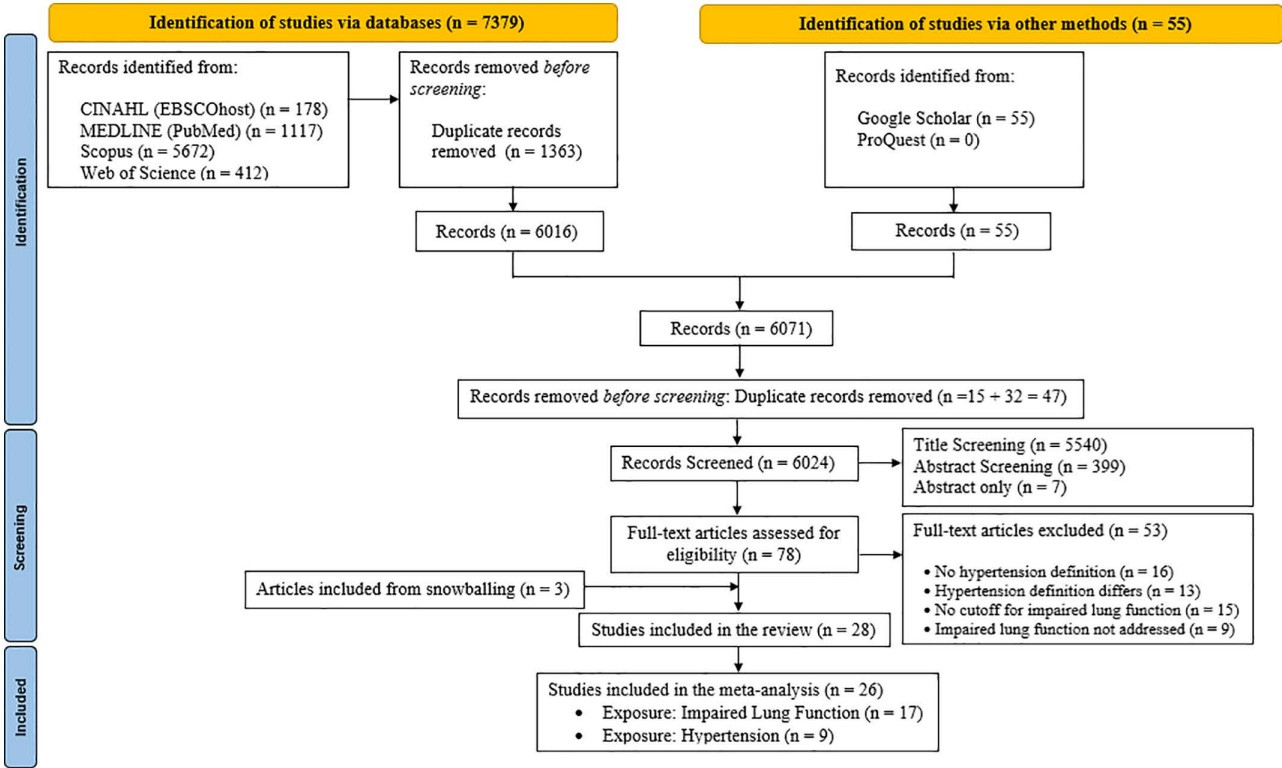

**Fig 1. PRISMA flow diagram of systematic review and meta-analysis.**

and abstracts of studies retrieved using the search strategy were screened independently by one reviewer (D.L.) to identify studies that meet the afore-mentioned inclusion criteria. The full text of the studies was independently assessed for eligibility by two reviewers (D.L. and A.K.). Disagreements between individual judgments were resolved through the discussion between reviewers.

## Data extraction

Data were extracted by using a data extraction table which was designed with the use of Microsoft Excel by one author (D.L.) and the details of the extraction were subsequently verified by two different independent authors (A.K. and M.M). Any disagreements were identified and resolved through discussion. The data extracted included authors, title, year of publication, study setting (region and country), study design, population, study period, inclusion and exclusion criteria of the study, sample size, gender, number of male and female individuals, mean age, mean body mass index and ethnicity of the participants, relevant variables, statistical method used, odds ratios (ORs) and 95% confidence intervals (CIs), exposure and the outcome, findings, and limitations. Prevalence of lung function impairment among hypertensive and non-hypertensives, and prevalence of hypertension among lung function impaired and normal lung function individuals were also extracted from the included studies. When a study reported more than one method of assessing the impaired lung function using the spirometric indices, effect sizes of different indices were extracted and analysed as separate studies. For studies where hypertension-lung function association was not the primary research question, we extracted relevant data from baseline characteristics tables.

Authors of relevant studies without the required data were contacted via e-mail to request sharing of unpublished data.

## Quality assessment

The full-text articles meeting eligibility were examined by two review authors (D.L. and A.K.) independently to assess the risk of bias. The Newcastle-Ottawa Scale (NOS) (S4 and S5 Files) or Joanna Briggs Institute (JBI) critical appraisal checklist (S6 File) were used as appropriate tools to assess the quality of included studies. The quality of the selected cohort and case-control studies was evaluated using the NOS, which is based on a star-rating systems and assigns scores ranging from 0 to 9 based on three domains: section, comparability and outcome/exposure with the maximum score of nine reflects the highest quality [44]. Articles with scores of 7 or above were rated as high quality, those with scores between 5 and 6 were considered average-quality, and those with scores of 4 or below were categorized as low quality [45].

For qualitative assessment of the cross-sectional studies, the JBI Critical Appraisal Checklist comprising 8 questions was used [46]. This checklist evaluates various aspects such as sample selection, subject description, exposure and subject condition measurement, identification and control of confounding factors, outcome assessment, and statistical analysis. Each item was rated as yes, no, unclear, or not applicable. If the answer was yes, the question was assigned a score of 1, whereas a score of 0 was assigned when the answer indicated no, unclear, or not applicable. Studies scoring 7 or above are deemed to have a low risk of bias and high quality, scores between 4 and 6 indicate a moderate risk or moderate quality, and scores below 4 suggest a high risk of bias or low quality [45]. Any disagreements were identified and resolved through discussion.

## Statistical analysis

Odds ratios (ORs) were extracted where available, or odds ratios were calculated from reported data or raw data received from authors to express the association between hypertension and impaired lung function and vice versa. The extracted raw data was used to compute the pooled odds ratio (OR) estimates with 95% confidence interval (CI) to assess the strength of the association between hypertension and impaired lung function as well as the association between impaired lung function and hypertension.

Studies were separately analyzed based on hypertension as the exposure and impaired lung function as the outcome, and vice versa. Initial meta-analyses were conducted using unadjusted ORs where available in the selected observational studies to identify whether there is a significant bidirectional association between hypertension and impaired lung function and followed up with the meta-analyses of the adjusted ORs obtained from the selected studies.

When a study has been conducted in several sites and if the site level information is available, those were included in the meta-analyses as different samples.

Considering the inter-study variations including differences in study design, potential differences in study populations and methodologies of retrieved studies meta-analyses were conducted to estimate the pooled effect size using the random-effect model. Forest plots were generated for each meta-analysis to illustrate the effect sizes and confidence intervals of each study and the pooled effect size.

The heterogeneity across the included studies was measured using Cochran's Q test and the statistical heterogeneity was quantified using $I^2$ statistic, in which an $I^2$ value of 25%, 50%, and 75% suggesting a low, moderate, and high degree of heterogeneity respectively [47,48]. Unless otherwise specified, outcomes were deemed statistically significant at $p < 0.05$.

To determine potential sources of heterogeneity subgroup analyses were performed by stratifying studies according to the study design, lung function impairment type, the country where the study was conducted, the geographical continent of study and adjustment relevant confounding factors (yes or no) to determine how the association between hypertension and impaired lung function and vice versa differ in terms of the above stratified variables.

To further detect the sources of heterogeneity, if a minimum of 10 studies were pooled a meta-regression was conducted considering the adjustments done for the following potential confounding variables: age, sex, smoking, Body Mass Index (BMI), education, race (black or white), physical inactivity, alcohol consumption levels, diabetes status, dyslipidemia,

household income level, waist circumference, obesity, residence, C-reactive protein (CRP) and macronutrient intake (protein carbohydrate, and total fat). Due to the limited number of studies within the analyses, the meta-regression was only conducted during the analysis of impaired lung function as the exposure and hypertension as the outcome.

Potential publication bias was graphically identified using funnel plots and assessed through the Egger's test to quantify any publication biases captured by the funnel plots. If significant publication bias was detected, a trim-and-fill method was performed to examine whether studies should be imputed to make the results more symmetrical to minimize the effects of publication bias [49,50].

"Leave-one-out" analysis was used to assess whether any one study has a dominant effect on the outcomes. Further, based on the quality assessment, sensitivity analysis was performed by excluding one study at a time and assessing whether the results are strongly influenced by a single study in order to exclude poor-quality studies. Studies with NOS quality rating with 4 or below and JBI score below 4 were removed and the meta-analyses were repeated to see if the association stood still. All statistical analysis were conducted using R Studio software (R 4.5.1; R Foundation) using the meta package [51].

## Results

### Selection of studies

A total of 7434 studies were identified during the initial search, with 7379 found through databases and 55 identified through other methods (Fig 1). After removing 1410 duplicates, 6024 unique records remained. These were screened based on titles and abstracts, leading to the exclusion of 5946 studies that did not meet the inclusion criteria. A total of 78 full-text articles were assessed for eligibility, of which 53 were excluded for reasons such as no definition of hypertension (n = 16), differing hypertension definitions (n = 13), no cutoff for impaired lung function (n = 15), and failure to address impaired lung function (n = 9), ultimately including 28 articles in the review (Table 1). Based on the availability of the required odds ratios, only 26 articles were included in the meta-analysis.

### Study characteristics

The summary of the study characteristics of the included studies in the systematic review can be found in Table 1 and Table 2. There were 22 cross-sectional studies [18,24,27,33,35–39,41,42,52–62], 5 case-control studies [16,32,34,63,64] and 1 cohort study [40]. Of these, 17 studies examined impaired lung function as exposure and hypertension as the outcome, while 11 studies investigated hypertension as the exposure and impaired lung function as the outcome.

### Exposure as impaired lung function and outcome as hypertension

The 17 studies included in this study consisted of 14 cross-sectional, 2 case-control and 1 cohort study spanning for a period of 30 years (1989–2019) comprised a total of 134823 participants, of whom 51.4% were female. Mean age across many included studies fell between 30 and 60 years (n = 10/17, 58.8%). Four studies were conducted in the United States; three in Korea; three studies combined countries globally; and the remaining studies were conducted in Australia, Austria, Brazil, Iran, Italy, Poland, and Sweeden. Two studies comprised an exclusively male population and the rest included both males and females. The type of lung function impairment assessed varied among the included studies. Specifically, nine studies focused only on obstructive impairment, two studies only on restrictive impairment, five studies on both obstructive and restrictive lung impairments, and one study assessed obstructive, restrictive, and mixed lung impairments (Table 1).

### Exposure as hypertension and outcome as impaired lung function

The 11 studies had a combined total of 13332 participants, with data collection occurring between 1989 and 2018. This included 8 cross-sectional studies and 3 case-control studies. The percentage of female participants in the included studies ranged from 43.0% to 53.8%. Data on female proportion were missing for some studies and were not included in

**Table 1. Characteristics of the 17 included observational studies with hypertension as the outcome.**

| Study (Author, year) | Continent | Country | Study Design | Study Period | Sample Size | Male (%) | Age (Years) Mean±SD | Adjusted Factors | Lung function impairment type | Major Findings |
|---|---|---|---|---|---|---|---|---|---|---|
| Bozek et al., 2016 | Europe | Poland | CCS | 2013 - 2014 | 3183 | 53.7 | 67.43 ± 5.7 | | Obstructive | Asthmatics had a significantly higher prevalence of HT. |
| Dharmage et al., 2023 | Australia | Australia | CSS | 2012–2016 | 2422 | 48.47 | | | Restrictive Obstructive Mixed | Individuals with the restrictive only pattern had the significantly highest prevalence of HT. |
| Di Raimondo et al., 2020 | Europe | Italy | CCS | 2018 - 2019 | 80 | 37.5 | 52.1 ± 12.5 | Age, sex, diabetes, fasting glucose, BMI | Obstructive | Once adjusted, asthma (OR = 3.66, 95% CI: 1.29–11.1, p=0.008), and severe asthma (OR = 4.32, 95% CI:1.88–9.54, p<0.001), were independently and significantly associated with an increased risk of HT. |
| Ferguson et al., 2014 | North America | USA | CSS | May 2004 – April 2006 and July 2007 December 2009 | 812 | 33 | 46 ± 14 | History of OSA, Age, sex, BMI, Smoking, asthma related variables | Obstructive | HT was significantly associated with lower $FEV_1$% (particularly 60–69%) and OSA, with stronger associations observed in combined OSA categories. |
| Jo et al., 2015 | Asia | Korea | CSS | 2010 - 2012 | 4057 | 100 | 56.89 ± 10.2 | Age | Obstructive | HT was significantly more prevalent in the COPD group compared to the non-COPD group in age-adjusted analyses. |
| Kaufmann et al., 2024 | Europe | Austria | CSS | 2011 | 9466 | 46.5 | | | Obstructive | Arterial HT was significantly more common in individuals with AL than in those with normal lung function. |
| Kiani & Ahmadi, 2021 | Asia | Iran | CSS | 2015 - 2019 | 6961 | 47.4 | 49.44 ± 9.29 | Age, sex, education level, BMI, smoking status | Obstructive | A significant association was found between obstructive lung impairment and HT, with a 30.2% prevalence in COPD patients and an adjusted OR of 1.4 (95% CI: 1.02–1.99) compared to healthy individuals. |

*(Continued)*

| Study (Author, year) | Continent | Country | Study Design | Study Period | Sample Size | Male (%) | Age (Years) Mean±SD | Adjusted Factors | Lung function impairment type | Major Findings |
|---|---|---|---|---|---|---|---|---|---|---|
| Kim et al., 2017 | Asia | Korea | CSS | 2010–2012 | 4043 | 100 | 57.05 ± 10.7 | Age, BMI, smoking status, diabetes, metabolic syndrome, stroke | Obstructive | COPD was significantly and independently associated with HT (adjusted OR = 1.71, 95% CI: 1.37–2.13, p < 0.001). The association between COPD and HT remained significant even after adjusting for age and smoking. |
| Kulbacka-Ortiz et al., 2022 | Asia, Africa, Europe, North America, Australia | Turkey, Algeria, Norway, Malawi, SA, Kyrgyztan, Sri Lanka, Benin, Morocco, China, Germany, Poland, USA, Portugal, England, NL, Philippines, India, Malaysia, Iceland, Saudi Arabia, Austria, Tunisia, Australia, Estonia, Albania, Sweden, Canada | CSS | | 22764 | 47.9 | 55.05 ± 10.4 | Sex, age, BMI, smoking status, pack-years, education | Restrictive | After adjusting for age, sex, education, BMI, and smoking, HT was significantly associated with a higher likelihood of restricted spirometry (OR = 1.50, 95% CI: 1.39–1.63), with females showing a stronger association. |
| Lindberg et al., 2011 | Europe | Sweden | CSS | 2002 - 2004 | 1986 | 54.6 | 64.55 | Gender, age, BMI, smoking habits | Restrictive Obstructive | COPD stage I was not significantly associated with HT (OR = 1.26, 95% CI: 0.99–1.61). COPD stage ≥II was **significantly** associated with increased risk of HT (**OR = 1.41**, 95% CI: 1.08–1.61). RLF was **not significantly** associated with HT (**OR = 1.36**, 95% CI: 0.98–1.89). |
| Mannino et al., 2008 | North America | USA | Cohort Study | 1989 - 1990 5 years of Follow up | 20296 | 44.5 | | Age, sex, race, smoking status, education level, BMI. | Restrictive Obstructive | GOLD stage 2, 3, or 4 COPD, as well as RLF, were significantly associated with a higher prevalence of HT (OR =1.4, 95% CI:1.2–1.7 for GOLD 2; OR = 1.6, 95% CI: 1.3–1.9 for GOLD 3/4; and OR = 1.3, 95% CI: 1.1–1.6 for RLF), while GOLD stage 1 was not significant. |

*(Continued)*

**Table 1.** (Continued)

| Study (Author, year) | Continent | Country | Study Design | Study Period | Sample Size | Male (%) | Age (Years) Mean±SD | Adjusted Factors | Lung function impairment type | Major Findings |
|---|---|---|---|---|---|---|---|---|---|---|
| Mannino et al.2012 | East Asia, West Asia, Central Europe, Northern Europe, North America, Southeast Asia, Australia | China, Turkey, Austria, SA, Iceland, Germany, Poland, Norway, Canada, USA, The Philippines, Australia, UK, Sweden | CSS | 2008 | 9762 | 47.8 | | Age, smoking status, BMI, site (Adjusted when the exposure is restricted and obstructed separately) | Restrictive Obstructive | Both obstructive and RLF impairments were significantly associated with a higher prevalence of HT, with adjusted OR of 2.08 (95% CI: 1.80–2.41) for restricted spirometry and 1.44 (95% CI: 1.25–1.65) for obstructive spirometry, compared to normal lung function. |
| Methvin et al., 2009 | North America | USA | CSS | | 508 | 40.6 | | Age, sex, smoking status, education, and BMI | Restrictive Obstructive | HT was significantly associated with RLF, while a non-significant association was observed with OLF, possibly due to sample size limitations. |
| Park et al., 2015 | Asia | Korea | CSS | 2007 - 2012 | 16151 | 43.2 | 57.5 ± 11.1 | | Restrictive Obstructive | HT prevalence increases significantly with the severity of airway obstruction in both men and women. |
| Sperandio et al., 2016 | South America | Brazil | CSS | | 374 | 42.14 | 41 ± 14 | Age, gender, race, education, self-reported CVD risk factors (SAH, diabetes, dyslipidemia, smoking, obesity, and physical inactivity), body composition (fat body mass), peripheral muscle function (PTQ and PTB), postural balance (COP-EO), and cardiorespiratory fitness (peak $VO_2$). | Restrictive | HT was significantly associated with a restrictive lung pattern, even after adjusting for confounders, with an OR of 17.5 (95% CI: 1.65–184.8). |

*(Continued)*

**Table 1.**  (Continued)

| Study (Author, year) | Continent | Country | Study Design | Study Period | Sample Size | Male (%) | Age (Years) Mean±SD | Adjusted Factors | Lung function impairment type | Major Findings | |
|---|---|---|---|---|---|---|---|---|---|---|---|
| Triest et al., 2019 | Asia, Africa, Europe, North America, Australia | Turkey, Algeria, Norway, Malawi SA, Kyrgyztan Sri Lanka, Benin, Morocco, China, Germany, Poland, USA, Portugal, England, NL, Philippines, India, Malaysia, Iceland, Saudi Arabia, Austria, Tunisia, Australia, Estonia, Albania, Sweden, Canada | CSS | As of November 2016 | 22764 | 47.9 | 55.05 ± 10.4 | Age, sex, smoking (including pack-years), BMI, education | Obstructive | Significant unadjusted association; not significant after adjustment. Age and smoking explain most of the association. | |
| Yang et al., 2020 | North America | USA | CSS | 2007–2012 | 13237 | 50.5 | 46.89 ± 15.6 | Age, gender, race, BMI, education level, income level, smoking status, alcohol use, macronutrient intake (protein, carbohydrate, total fat), EOS%, NEU%, HT, DM and dyslipidemia | Obstructive | In COPD patients, decreased $FEV_1\%$ and FVC% predicted were significantly associated with higher ORs of HT, even after adjusting for confounders. | |

**Abbreviations:** AL: Airflow limitation; CCS: Case-Control Study; CSS: Cross-sectional study; BMI: Body Mass Index; OR: Odds Ratio; CI: Confidence Interval; OSA: Obstructive Sleep Apnea; $FEV_1\%$: Percent predicted Forced Expiratory Volume in one second; COPD: Chronic Obstructive Pulmonary Disease; SA: South Africa; USA: United States of America; NL: Netherlands; RLF: Restricted lung function; UK: United Kingdom; CVD: Cardiovascular diseases; SAH: Systemic Arterial Hypertension; $VO_2$: Oxygen uptake; COP-EO: Center of pressure – eyes open COP area while standing with eyes open; PTQ: peak Torque of quadriceps; PTB: Peak Torque of biceps; EOS%: Eosinophils percent; NEU%: Neutrophils percent; DM: Diabetes Mellitus; HT: Hypertension; FVC%: Percent predicted Forced Vital Capacity; OLF: Obstructive Lung Function; GOLD: Global Initiative for Chronic Obstructive Lung Disease

this range. Mean age across many included studies fell between 30 and 60 years (n = 8/11, 72.7%). Among the selected studies, three studies were carried out in India, while two others were conducted in Korea. The remaining studies took place in several other countries worldwide (Ethiopia, Iceland, Italy, Japan, Nigeria, and United States). The assessment of lung function impairment varied across the selected studies, with 2 studies focusing on restrictive type, 5 on obstructive type, 3 on both obstructive and restrictive impairments, and 1 addressing obstructive, restrictive and mixed lung function impairments (Table 2).

## Quality assessment

When the risk of bias of included studies was evaluated using the NOS scale, for the studies investigating the impaired lung function as the exposure, one study indicated as high quality [40], whereas two studies were of average quality [63,64]. When using the JBI Critical Appraisal Checklist, 2 studies demonstrated a low risk of bias while 10 studies were identified to have a moderate risk of bias. Two studies indicated a high risk of bias. The risk of bias in included observational studies can be found in Tables 3 and 4.

**Table 2. Characteristics of the 11 included observational studies with impaired lung function as the outcome.**

| Study (Author, year) | Continent | Country | Study Design | Study Period | Sample Size | Male (%) | Age (Years) Mean ± SD | Adjusted Factors | Lung function impairment type | Major Findings |
|---|---|---|---|---|---|---|---|---|---|---|
| Birhan & Abebe, 2018 | Africa | Ethiopia | CSS | 2016 | 122 | 47.5 | 51 ± 7.1 | | Restrictive | Hypertensive patients exhibited significantly lower pulmonary function values (FVC, $FEV_1$) compared to controls, with a higher $FEV_1$% indicating a dominant restrictive pulmonary defect, while no significant association was found between pulmonary function and the severity of HT. |
| Femi-Adeoye et al., 2024 | Africa | Nigeria | CCS | 1st of May and the 21st of July 2017 | 210 | 46.2 | 50.2 ± 6.0 | Age, WC, BMI | Restrictive Obstructive | Poor lung function was significantly more prevalent among hypertensives (33.3%) than normotensives (21%) (p = 0.044) and restriction pattern was the predominant lung function abnormality. After adjusting for these factors, respondents with high blood pressure in the hypertensive group were about three times more likely to have poor lung function (OR = 2.995, 95% CI: 1.243–7.171) |
| Ferrari et al., 2019 | Europe | Italy | CCS | 2006 and 2010 2007 - 2015 | 2463 | 49.3 | 49.5 ± 12.8 | Sex, age, education, smoking habits, alcohol consumption, physical activity, BMI, comorbidity indicators (diabetes, HT, dyslipidemia | Obstructive | HT was significantly associated with asthma (RRR = 1.41, 95% CI: 1.06–1.89). |
| Kim et al., 2021 | East Asia | Korea | CSS | 2016–2018 | 3195 | 50.5 | 45.09 ± 4.3 | Age, sex, BMI, alcohol consumption, smoking status, educational level, household income level, residence | Obstructive | Presence of HT was significantly positively associated with early COPD. |
| Lee et al., 2020 | Asia | South Korea | CSS | 2014 - 2015 | 4644 | 32.1 | 54.01 ± 10.21 | Age, sex, BMI, WC, lifetime smoking, alcohol consumption, aerobic physical activity engagement, HGS | Restrictive | HT with normal FPG was associated with reduced lung function, though the adjusted OR of 1.45 (95% CI: 0.91–2.33) did not reach statistical significance. |
| Margretardottir et al., 2009 | Europe | Iceland | CSS | 2004 - 2005 | 750 | 53.07 | 56.40 ± 10.9 | HT, BMI, CRP, age, sex, current smoking, pack years | Obstructive | HT was significantly associated with lower $FEV_1$% and FVC%, especially in men, and this association remained significant after adjusting for age, sex, smoking, and pack-years. |
| Patil et al., 2012 | Asia | India | CCS | 2007 - 2008 | 200 | 57 | 49.53 ± 9.8 | | Obstructive | The prevalence of OAD was significantly higher in hypertensive individuals (16.66%), with HT being independently associated with reduced lung function ($FEV_1$/FVC < 70%) and showing an OR of 8.044, highlighting a strong link between HT and IPF. |
| Shah et al., 2014 | Asia | India | CSS | | 60 | | 54.39 ± 8.4 | | Restrictive Obstructive | Patients with HT demonstrated significantly lower pulmonary function test parameters compared to controls (p < 0.05), with a predominantly obstructive pattern observed in 74% of cases. |

*(Continued)*

**Table 2.** (Continued)

| Study (Author, year) | Conti-nent | Coun-try | Study Design | Study Period | Sam-ple Size | Male (%) | Age (Years) Mean ± SD | Adjusted Factors | Lung function impair-ment type | Major Findings |
|---|---|---|---|---|---|---|---|---|---|---|
| Taneda et al., 2004 | North Amer-ica | United States | CSS | 1989–1994 | 678 | 53.2 | | Pulse wave velocity, sex, age, BMI, cholesterol ratio, HT status, diabetes status, drinking status, smoking status | Restric-tive Obstruc-tive | HT is significantly positively associated with abnormal FVC% and FEV$_1$%. |
| Utsugi et al., 2016 | Asia | Japan | CSS | 2010–2012 | 950 | 49.5 | 64.9 ± 8.7 | Age, lifelong cigarette con-sumption, and the presence of HT or dyslipidemia, | Obstruc-tive | HT tended to be associated with more severe airflow limitation (GOLD II/III), but this was not significant (p = 0.09). Presence of HT was a risk factor for airflow limitation (14.6%) |
| Yadav et al., 2015 | Asia | India | CCS | | 60 | | | | Restric-tive Obstruc-tive Mixed | An inverse relation is found between HT and pulmonary functions predomi-nantly restrictive type of pattern. While non-significant effects are observed with severity of illness. |

**Abbreviations:** CCS: Case-Control Study; CSS: Cross-sectional study; FVC: Forced Vital Capacity; FEV$_1$: Forced Expiratory Volume in one second; FEV$_1$%: Percent predicted Forced Expiratory Volume in one second; FVC%: Percent predicted Forced Vital Capacity; HT: Hypertension; RRR: Relative Risk Ratio; BMI: Body Mass Index; COPD: Chronic Obstructive Pulmonary Disease; WC: Waist Circumference; FPG: Fasting plasma glucose; OR: Odds Ratio; CI: Confidence Interval; OAD: Obstructive Airway Disease; IPF: Impaired pulmonary function; GOLD: Global Initiative for Chronic Obstruc-tive Lung Disease; HGS: Handgrip Strength; CRP: C-reactive protein

**Table 3. Risk of bias summary for the cohort and case-control studies with impaired lung function as the exposure.**

| Study (Author, year) | Study Design | Selection | Comparability | Exposure/Outcome | Total Score |
|---|---|---|---|---|---|
| Bozek et al., 2016 [63] | Case Control Study | 4 | 0 | 2 | 6 |
| Di Raimondo et al., 2020 [64] | Case Control Study | 2 | 2 | 2 | 6 |
| Mannino et al., 2008 [40] | Cohort Study | 2 | 2 | 3 | 7 |

**Table 4. Risk of bias summary for the cross-sectional studies with impaired lung function as the exposure.**

| Study | Quality Score |
|---|---|
| Dharmage et al., 2023 [52] | 4 |
| Ferguson et al., 2014 [42] | 6 |
| Jo et al., 2015 [54] | 6 |
| Kaufmann et al., 2024 [55] | 2 |
| Kiani & Ahmadi., 2021 [56] | 6 |
| Kim et al., 2017 [65] | 7 |
| Kulbacka-Ortiz et al., 2022 [18] | 6 |
| Lindberg et al., 2011 [41] | 6 |
| Mannino et al., 2012 [58] | 5 |
| Methvin et al., 2009 [59] | 5 |
| Park et al., 2015 [60] | 3 |
| Sperandio et al., 2016 [61] | 6 |
| Triest et al., 2019 [27] | 7 |
| Yang et al., 2020 [62] | 6 |

For the studies investigating hypertension as the exposure, two studies were of high quality, and one study was of average quality when evaluated using the NOS. According to the JBI checklist there were 3 studies with low risk of bias, 3 studies were with moderate risk of bias and two were having high risk of bias. The risk of bias in the included observational studies can be found in Table 5 and Table 6.

### Association between hypertension and impaired lung function

**Impaired lung function as the exposure and hypertension as the outcome.** 31 studies (17 original papers) reported the relationship between impaired lung function and hypertension (n = 192055) when using unadjusted odds ratios. Among those 23 studies consistently revealed a significantly higher association between impaired lung function and hypertension and the remaining 8 studies reported no significant relationship between impaired lung function and hypertension (Fig 2). Overall, the summary estimate demonstrated that patients with impaired lung function had significantly higher risk of having hypertension compared with normal lung functioned individuals (unadjusted OR = 1.6957; 95% CI: [1.5312–1.8779], $p < 0.0001$). The heterogeneity across the studies was significant ($I^2 = 84.2\%$, $p < 0.0001$).

### Publication bias

Although the visual inspection of the funnel plot shows a little asymmetry (Figs 3 and 4), the results of the Egger's test indicated that there is no significant asymmetry ($t = -0.22$, $df = 24$, $p = 0.8272$), suggesting no evidence of small-study effects. Five studies were omitted from the test, as the SE variable was missing.

Of the 31 studies initially considered, five studies were excluded from the trim-and-fill method due to missing standard errors. The remaining 26 studies were included, and no additional studies were imputed (Fig 5) suggesting that there is no evidence of publication bias. The pooled effect size was 0.5145 [95% CI: 0.4033; 0.6257], $p < 0.0001$. $I^2 = 86.2$ (substantial variability).

86 studies (13 original papers) reported the relationship between impaired lung function and hypertension (n = 181371) when using adjusted odds ratios. Among those 31 studies consistently revealed a significantly higher association between

**Table 5. Risk of bias summary for the case-control studies with hypertension as the exposure.**

| Study (Author, year) | Study Design | Selection | Comparability | Exposure | Total Score |
|---|---|---|---|---|---|
| Ferrari et al., 2019 [34] | Case Control Study | 4 | 2 | 1 | 7 |
| Patil et al., 2012 [16] | Case Control Study | 3 | 1 | 2 | 6 |
| Yadav et al., 2015 [32] | Case Control Study | 3 | 2 | 2 | 7 |

**Table 6. Risk of bias summary for the cross-sectional studies with hypertension as the exposure.**

| Study | Quality Score |
|---|---|
| Birhan & Abebe., 2018 [33] | 2 |
| Femi-Adeoye et al., 2024 [53] | 8 |
| Kim et al., 2021 [35] | 7 |
| Lee et al., 2020 [36] | 8 |
| Margretardottir et al., 2009 [24] | 6 |
| Shah et al., 2014 [37] | 3 |
| Taneda et al., 2004 [38] | 5 |
| Utsugi et al., 2016 [39] | 5 |

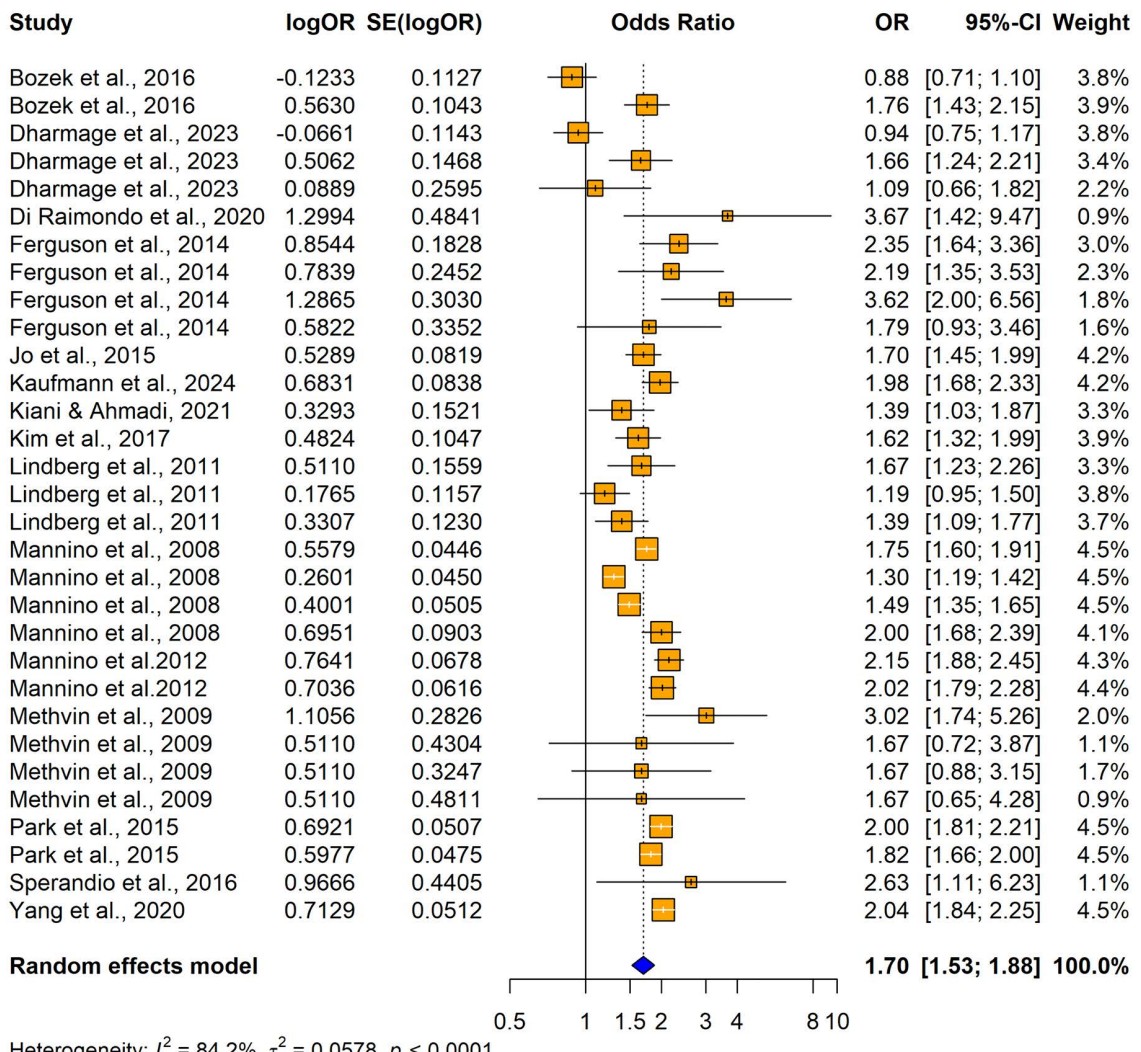

**Fig 2. Forest plot showing the association between impaired lung function and hypertension among adults when using impaired lung function as the exposure and hypertension as the outcome (using unadjusted odds ratios).**

impaired lung function and hypertension and the remaining 55 studies reported no significant relationship between impaired lung function and hypertension (Fig 6). Overall, the summary estimate demonstrated that patients with lung function impairment had significantly higher risk of having hypertension compared with normal lung functioned individuals (adjusted OR = 1.3986; 95% CI: [1.3107–1.4925], p < 0.0001). It was found out that there was a significant substantial heterogeneity across the studies. ($I^2$ = 59.8%, p < 0.0001).

Both visual inspection of the funnel plot (Figs 7 and 8) and the results of the Egger's test for publication bias indicated that there is no significant asymmetry (t = 0.8403, df = 84, p = 0.4031), suggesting no evidence of small-study effects.

Although the Egger's test and visual inspection of the funnel plot indicated no significant publication bias, seventeen studies were imputed during the trim-and-fill method to correct for slight asymmetry (Fig 9). While these results suggest that small-study effects cannot be completely ruled out, the adjusted pooled effect size was only slightly reduced to 0.1622 [95% CI: 0.0819; 0.2425], p < 0.0001, with $I^2$ = 36.5% (moderate variability), and the direction and statistical significance of the association remained consistent with the original findings.

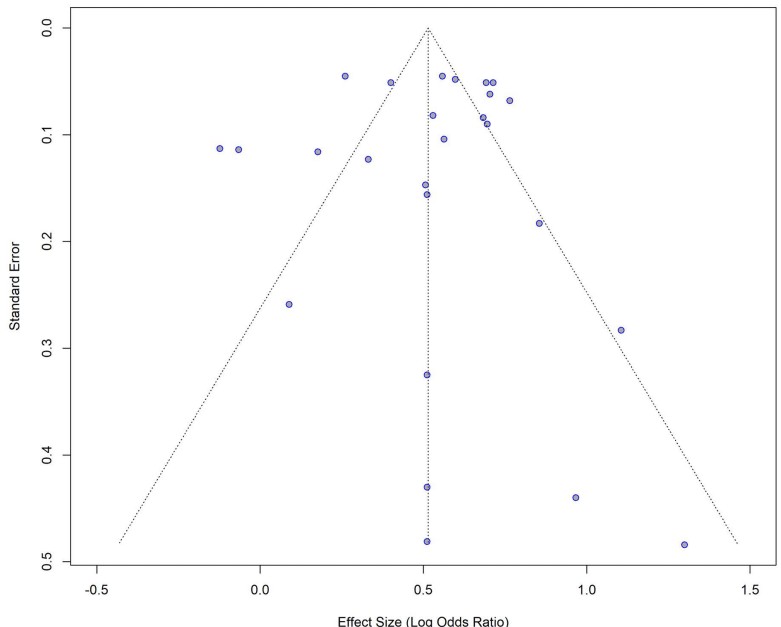

**Fig 3. Funnel plot for the studies with crude (unadjusted) ORs for impaired lung function as the exposure and hypertension as the outcome.**

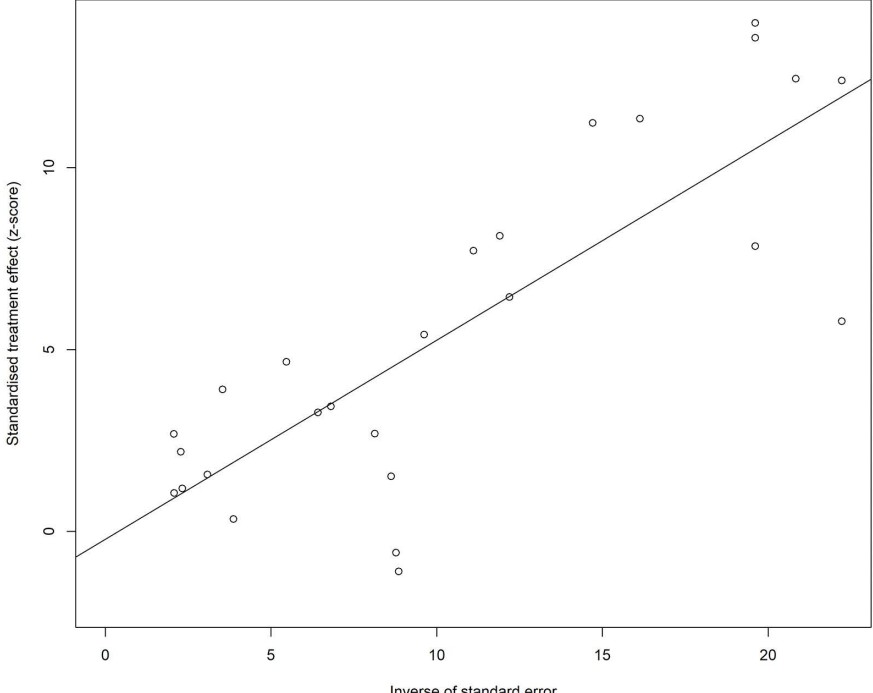

**Fig 4. Funnel plot with the regression line for the studies with crude (unadjusted) ORs for impaired lung function as the exposure and hypertension as the outcome.** The points are not symmetrically distributed around the regression line.

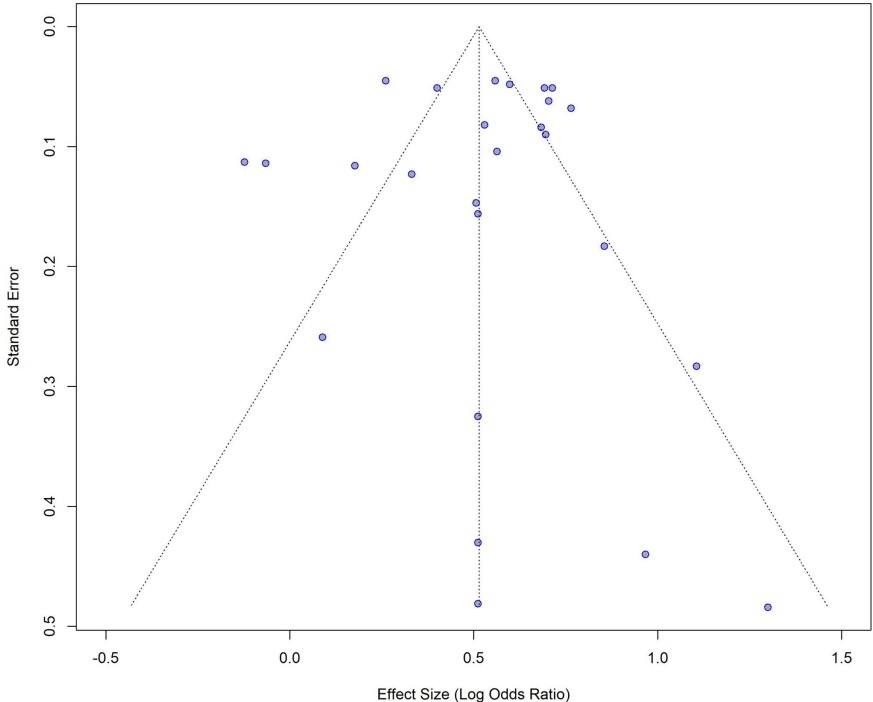

**Fig 5. Trim-and-fill funnel plot for the studies with crude (unadjusted) ORs for impaired lung function as the exposure and hypertension as the outcome.**

### Hypertension as the exposure and impaired lung function as the outcome

14 studies (9 original papers) reported the relationship between hypertension and impaired lung function (n = 13740) when using unadjusted odds ratios. Among those 10 studies consistently revealed a significantly higher association between hypertension and impaired lung function and the remaining 4 studies reported no significant relationship between hypertension and impaired lung function (Fig 10). Overall, the summary estimate demonstrated that patients with hypertension had significantly higher risk of having lung function impairment compared with normotensives (unadjusted OR = 2.9960; 95% CI: [1.8637–4.8161], p < 0.0001). The heterogeneity across the studies was significant ($I^2$ = 84.7%, p < 0.0001).

### Publication bias

Visual inspection of the funnel plot (Figs 11 and 12) and Egger's test for publication bias both indicated significant asymmetry (t = 3.72, df = 12, p = 0.0029), suggesting evidence of publication bias or small-study effects in the meta-analysis.

Seven studies were imputed using the trim-and-fill method to correct for asymmetry in the funnel plot (Fig 13). After imputation the adjusted effect size was 0.4201 [95% CI: −0.3421; 1.1822], p = 0.2800. $I^2$ = 85.4% (substantial variability). After adjusting for potential publication bias, the crude association between hypertension and impaired lung function lost statistical significance, suggesting that small-study effects may have inflated the initial estimate.

6 studies reported the relationship between hypertension and impaired lung function (n = 10355) when using adjusted odds ratios. Among those 4 studies consistently revealed a significantly higher association between hypertension and impaired lung function and the remaining 2 studies reported no significant relationship between hypertension and impaired lung function (Fig 14). Overall, the summary estimate demonstrated that patients with hypertension had significantly higher risk of having lung function impairment compared with normotensives (adjusted OR = 1.9417; 95%

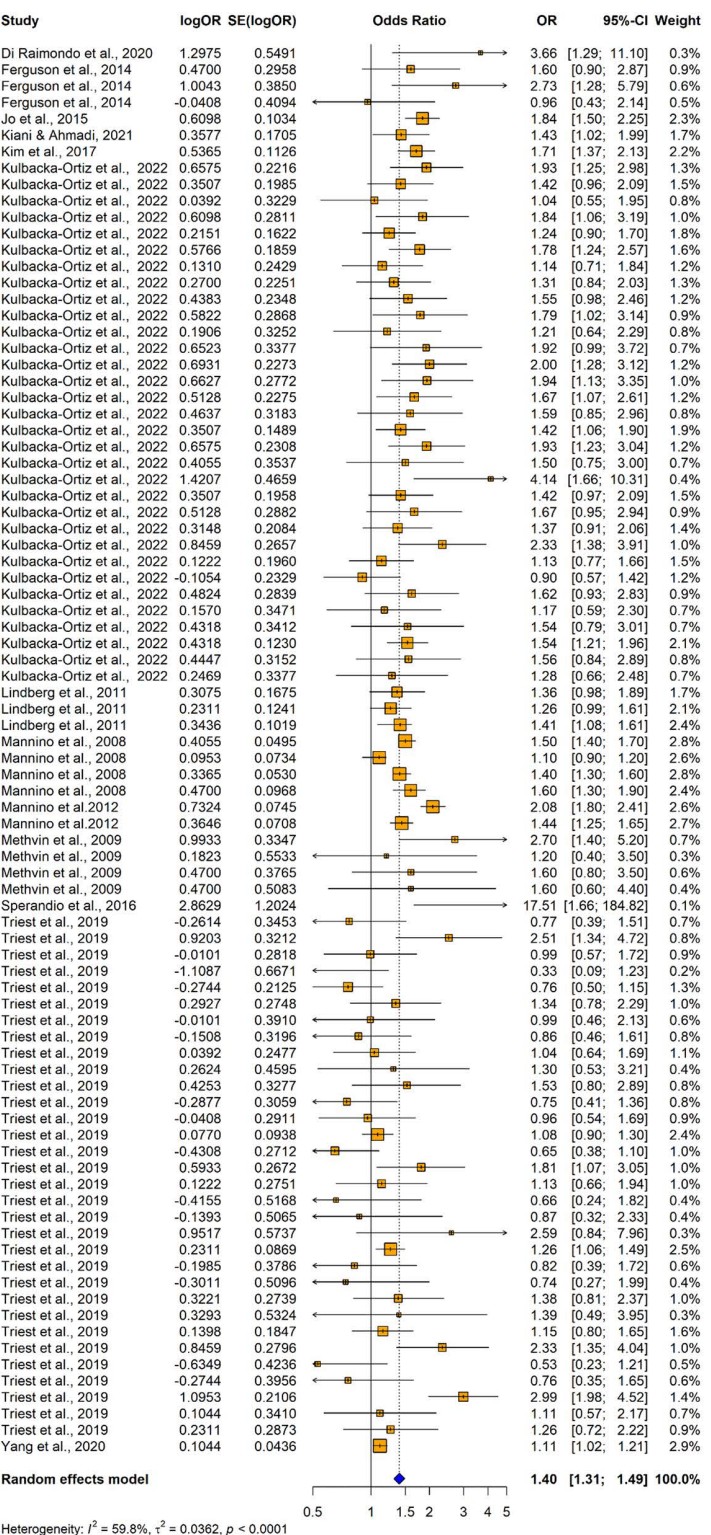

| Study | logOR | SE(logOR) | Odds Ratio | OR | 95%-CI | Weight |
|---|---|---|---|---|---|---|
| Di Raimondo et al., 2020 | 1.2975 | 0.5491 | | 3.66 | [1.29; 11.10] | 0.3% |
| Ferguson et al., 2014 | 0.4700 | 0.2958 | | 1.60 | [0.90; 2.87] | 0.9% |
| Ferguson et al., 2014 | 1.0043 | 0.3850 | | 2.73 | [1.28; 5.79] | 0.6% |
| Ferguson et al., 2014 | -0.0408 | 0.4094 | | 0.96 | [0.43; 2.14] | 0.5% |
| Jo et al., 2015 | 0.6098 | 0.1034 | | 1.84 | [1.50; 2.25] | 2.3% |
| Kiani & Ahmadi, 2021 | 0.3577 | 0.1705 | | 1.43 | [1.02; 1.99] | 1.7% |
| Kim et al., 2017 | 0.5365 | 0.1126 | | 1.71 | [1.37; 2.13] | 2.2% |
| Kulbacka-Ortiz et al., 2022 | 0.6575 | 0.2216 | | 1.93 | [1.25; 2.98] | 1.3% |
| Kulbacka-Ortiz et al., 2022 | 0.3507 | 0.1985 | | 1.42 | [0.96; 2.09] | 1.5% |
| Kulbacka-Ortiz et al., 2022 | 0.0392 | 0.3229 | | 1.04 | [0.55; 1.95] | 0.8% |
| Kulbacka-Ortiz et al., 2022 | 0.6098 | 0.2811 | | 1.84 | [1.06; 3.19] | 1.0% |
| Kulbacka-Ortiz et al., 2022 | 0.2151 | 0.1622 | | 1.24 | [0.90; 1.70] | 1.8% |
| Kulbacka-Ortiz et al., 2022 | 0.5766 | 0.1859 | | 1.78 | [1.24; 2.57] | 1.6% |
| Kulbacka-Ortiz et al., 2022 | 0.1310 | 0.2429 | | 1.14 | [0.71; 1.84] | 1.2% |
| Kulbacka-Ortiz et al., 2022 | 0.2700 | 0.2251 | | 1.31 | [0.84; 2.03] | 1.3% |
| Kulbacka-Ortiz et al., 2022 | 0.4383 | 0.2348 | | 1.55 | [0.98; 2.46] | 1.2% |
| Kulbacka-Ortiz et al., 2022 | 0.5822 | 0.2868 | | 1.79 | [1.02; 3.14] | 0.9% |
| Kulbacka-Ortiz et al., 2022 | 0.1906 | 0.3252 | | 1.21 | [0.64; 2.29] | 0.8% |
| Kulbacka-Ortiz et al., 2022 | 0.6523 | 0.3377 | | 1.92 | [0.99; 3.72] | 0.7% |
| Kulbacka-Ortiz et al., 2022 | 0.6931 | 0.2273 | | 2.00 | [1.28; 3.12] | 1.2% |
| Kulbacka-Ortiz et al., 2022 | 0.6627 | 0.2772 | | 1.94 | [1.13; 3.35] | 1.0% |
| Kulbacka-Ortiz et al., 2022 | 0.5128 | 0.2275 | | 1.67 | [1.07; 2.61] | 1.2% |
| Kulbacka-Ortiz et al., 2022 | 0.4637 | 0.3183 | | 1.59 | [0.85; 2.96] | 0.8% |
| Kulbacka-Ortiz et al., 2022 | 0.3507 | 0.1489 | | 1.42 | [1.06; 1.90] | 1.9% |
| Kulbacka-Ortiz et al., 2022 | 0.6575 | 0.2308 | | 1.93 | [1.23; 3.04] | 1.2% |
| Kulbacka-Ortiz et al., 2022 | 0.4055 | 0.3537 | | 1.50 | [0.75; 3.00] | 0.7% |
| Kulbacka-Ortiz et al., 2022 | 1.4207 | 0.4659 | | 4.14 | [1.66; 10.31] | 0.4% |
| Kulbacka-Ortiz et al., 2022 | 0.3507 | 0.1958 | | 1.42 | [0.97; 2.09] | 1.5% |
| Kulbacka-Ortiz et al., 2022 | 0.5128 | 0.2882 | | 1.67 | [0.95; 2.94] | 0.9% |
| Kulbacka-Ortiz et al., 2022 | 0.3148 | 0.2084 | | 1.37 | [0.91; 2.06] | 1.4% |
| Kulbacka-Ortiz et al., 2022 | 0.8459 | 0.2657 | | 2.33 | [1.38; 3.91] | 1.0% |
| Kulbacka-Ortiz et al., 2022 | 0.1222 | 0.1960 | | 1.13 | [0.77; 1.66] | 1.5% |
| Kulbacka-Ortiz et al., 2022 | -0.1054 | 0.2329 | | 0.90 | [0.57; 1.42] | 1.2% |
| Kulbacka-Ortiz et al., 2022 | 0.4824 | 0.2839 | | 1.62 | [0.93; 2.83] | 0.9% |
| Kulbacka-Ortiz et al., 2022 | 0.1570 | 0.3471 | | 1.17 | [0.59; 2.30] | 0.7% |
| Kulbacka-Ortiz et al., 2022 | 0.4318 | 0.3412 | | 1.54 | [0.79; 3.01] | 0.7% |
| Kulbacka-Ortiz et al., 2022 | 0.4318 | 0.1230 | | 1.54 | [1.21; 1.96] | 2.1% |
| Kulbacka-Ortiz et al., 2022 | 0.4447 | 0.3152 | | 1.56 | [0.84; 2.89] | 0.8% |
| Kulbacka-Ortiz et al., 2022 | 0.2469 | 0.3377 | | 1.28 | [0.66; 2.48] | 0.7% |
| Lindberg et al., 2011 | 0.3075 | 0.1675 | | 1.36 | [0.98; 1.89] | 1.7% |
| Lindberg et al., 2011 | 0.2311 | 0.1241 | | 1.26 | [0.99; 1.61] | 2.1% |
| Lindberg et al., 2011 | 0.3436 | 0.1019 | | 1.41 | [1.08; 1.61] | 2.4% |
| Mannino et al., 2008 | 0.4055 | 0.0495 | | 1.50 | [1.40; 1.70] | 2.8% |
| Mannino et al., 2008 | 0.0953 | 0.0734 | | 1.10 | [0.90; 1.20] | 2.6% |
| Mannino et al., 2008 | 0.3365 | 0.0530 | | 1.40 | [1.30; 1.60] | 2.8% |
| Mannino et al., 2008 | 0.4700 | 0.0968 | | 1.60 | [1.30; 1.90] | 2.4% |
| Mannino et al.2012 | 0.7324 | 0.0745 | | 2.08 | [1.80; 2.41] | 2.6% |
| Mannino et al.2012 | 0.3646 | 0.0708 | | 1.44 | [1.25; 1.65] | 2.7% |
| Methvin et al., 2009 | 0.9933 | 0.3347 | | 2.70 | [1.40; 5.20] | 0.7% |
| Methvin et al., 2009 | 0.1823 | 0.5533 | | 1.20 | [0.40; 3.50] | 0.3% |
| Methvin et al., 2009 | 0.4700 | 0.3765 | | 1.60 | [0.80; 3.50] | 0.6% |
| Methvin et al., 2009 | 0.4700 | 0.5083 | | 1.60 | [0.60; 4.40] | 0.4% |
| Sperandio et al., 2016 | 2.8629 | 1.2024 | | 17.51 | [1.66; 184.82] | 0.1% |
| Triest et al., 2019 | -0.2614 | 0.3453 | | 0.77 | [0.39; 1.51] | 0.7% |
| Triest et al., 2019 | 0.9203 | 0.3212 | | 2.51 | [1.34; 4.72] | 0.8% |
| Triest et al., 2019 | -0.0101 | 0.2818 | | 0.99 | [0.57; 1.72] | 0.9% |
| Triest et al., 2019 | -1.1087 | 0.6671 | | 0.33 | [0.09; 1.23] | 0.2% |
| Triest et al., 2019 | -0.2744 | 0.2125 | | 0.76 | [0.50; 1.15] | 1.3% |
| Triest et al., 2019 | 0.2927 | 0.2748 | | 1.34 | [0.78; 2.29] | 1.0% |
| Triest et al., 2019 | -0.0101 | 0.3910 | | 0.99 | [0.46; 2.13] | 0.6% |
| Triest et al., 2019 | -0.1508 | 0.3196 | | 0.86 | [0.46; 1.61] | 0.8% |
| Triest et al., 2019 | 0.0392 | 0.2477 | | 1.04 | [0.64; 1.69] | 1.1% |
| Triest et al., 2019 | 0.2624 | 0.4595 | | 1.30 | [0.53; 3.21] | 0.4% |
| Triest et al., 2019 | 0.4253 | 0.3277 | | 1.53 | [0.80; 2.89] | 0.8% |
| Triest et al., 2019 | -0.2877 | 0.3059 | | 0.75 | [0.41; 1.36] | 0.8% |
| Triest et al., 2019 | -0.0408 | 0.2911 | | 0.96 | [0.54; 1.69] | 0.9% |
| Triest et al., 2019 | 0.0770 | 0.0938 | | 1.08 | [0.90; 1.30] | 2.4% |
| Triest et al., 2019 | -0.4308 | 0.2712 | | 0.65 | [0.38; 1.10] | 1.0% |
| Triest et al., 2019 | 0.5933 | 0.2672 | | 1.81 | [1.07; 3.05] | 1.0% |
| Triest et al., 2019 | 0.1222 | 0.2751 | | 1.13 | [0.66; 1.94] | 1.0% |
| Triest et al., 2019 | -0.4155 | 0.5168 | | 0.66 | [0.24; 1.82] | 0.4% |
| Triest et al., 2019 | -0.1393 | 0.5065 | | 0.87 | [0.32; 2.33] | 0.4% |
| Triest et al., 2019 | 0.9517 | 0.5737 | | 2.59 | [0.84; 7.96] | 0.3% |
| Triest et al., 2019 | 0.2311 | 0.0869 | | 1.26 | [1.06; 1.49] | 2.5% |
| Triest et al., 2019 | -0.1985 | 0.3786 | | 0.82 | [0.39; 1.72] | 0.6% |
| Triest et al., 2019 | -0.3011 | 0.5096 | | 0.74 | [0.27; 1.99] | 0.4% |
| Triest et al., 2019 | 0.3221 | 0.2739 | | 1.38 | [0.81; 2.37] | 1.0% |
| Triest et al., 2019 | 0.3293 | 0.5324 | | 1.39 | [0.49; 3.95] | 0.3% |
| Triest et al., 2019 | 0.1398 | 0.1847 | | 1.15 | [0.80; 1.65] | 1.6% |
| Triest et al., 2019 | 0.8459 | 0.2796 | | 2.33 | [1.35; 4.04] | 1.0% |
| Triest et al., 2019 | -0.6349 | 0.4236 | | 0.53 | [0.23; 1.21] | 0.5% |
| Triest et al., 2019 | -0.2744 | 0.3956 | | 0.76 | [0.35; 1.65] | 0.6% |
| Triest et al., 2019 | 1.0953 | 0.2106 | | 2.99 | [1.98; 4.52] | 1.4% |
| Triest et al., 2019 | 0.1044 | 0.3410 | | 1.11 | [0.57; 2.17] | 0.7% |
| Triest et al., 2019 | 0.2311 | 0.2873 | | 1.26 | [0.72; 2.22] | 0.9% |
| Yang et al., 2020 | 0.1044 | 0.0436 | | 1.11 | [1.02; 1.21] | 2.9% |
| **Random effects model** | | | | **1.40** | **[1.31; 1.49]** | **100.0%** |

Heterogeneity: $I^2 = 59.8\%$, $\tau^2 = 0.0362$, $p < 0.0001$

**Fig 6. Forest plot showing the association between impaired lung function and hypertension among adults when using impaired lung function as the exposure and hypertension as the outcome (using adjusted odds ratios).**

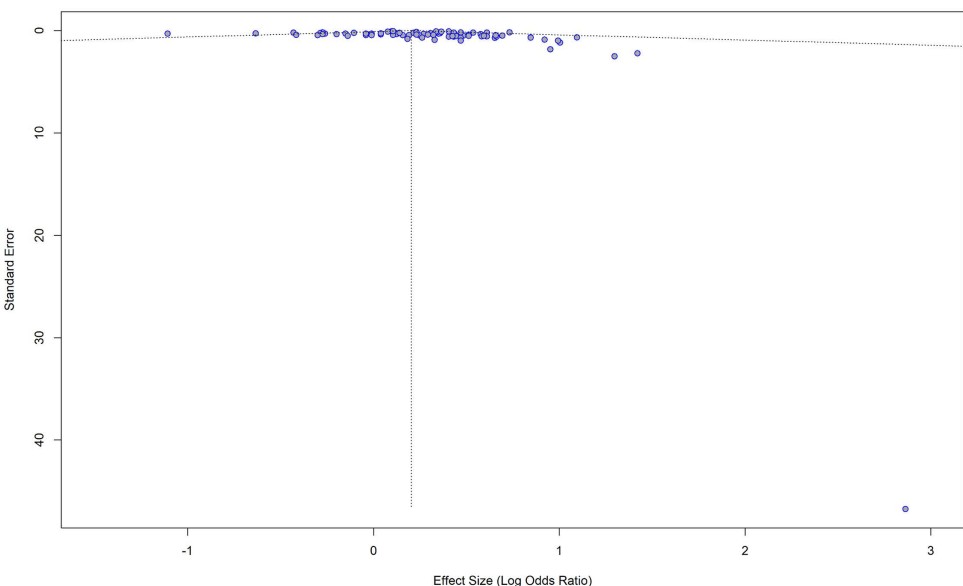

**Fig 7. Funnel plot for the studies with adjusted ORs for impaired lung function as the exposure and hypertension as the outcome.**

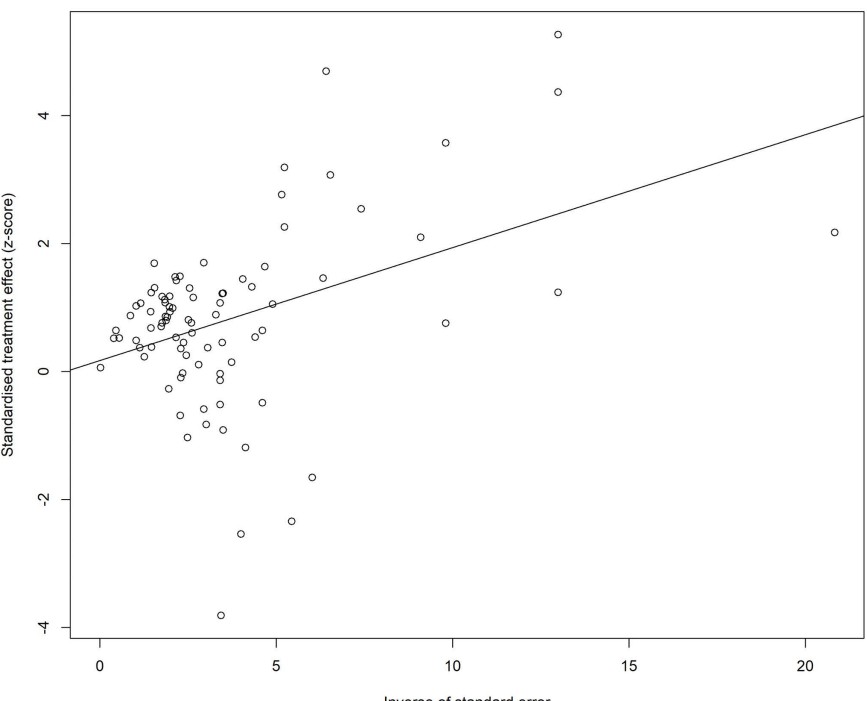

**Fig 8. Funnel plot with the regression line for the studies with adjusted ORs for impaired lung function as the exposure and hypertension as the outcome.** The points are symmetrically distributed around the regression line.

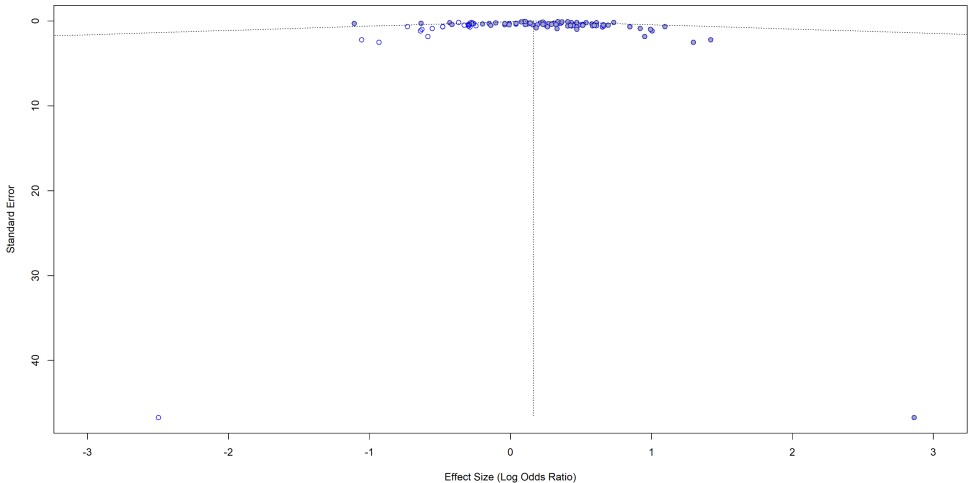

**Fig 9. Trim-and-fill funnel plot for the studies with adjusted ORs for impaired lung function as the exposure and hypertension as the outcome.**

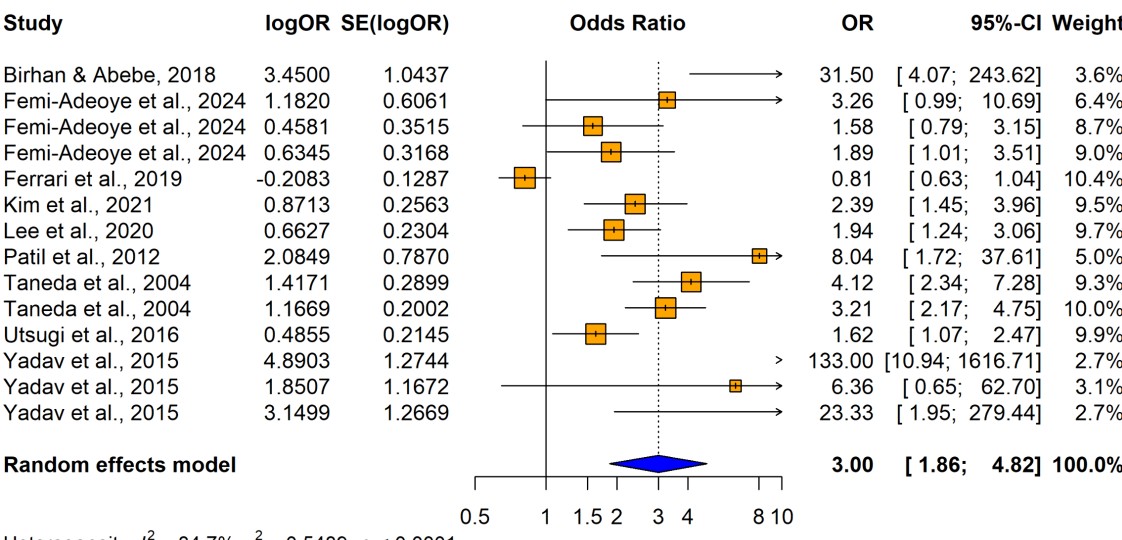

**Fig 10. Forest plot showing the association between hypertension and impaired lung function among adults when using hypertension as the exposure (using unadjusted odds ratios).**

CI: [1.5110–2.4953], p < 0.0001]. It was found out that there was no significant heterogeneity across the studies ($I^2 = 0.0\%$, p = 0.5992). A summary table comparing both directions of the associations between impaired lung function and hypertension is shown in Table 7.

The visual inspection of the funnel plot indicated some evidence of publication bias in this meta-analysis based on the asymmetry of the funnel plot and one study was omitted due to missing adjusted SE. (Fig 15). The number of studies (k = 4) was too small to test for small study effects (k.min = 10). Therefore, we refrained from performing Egger's test at this stage.

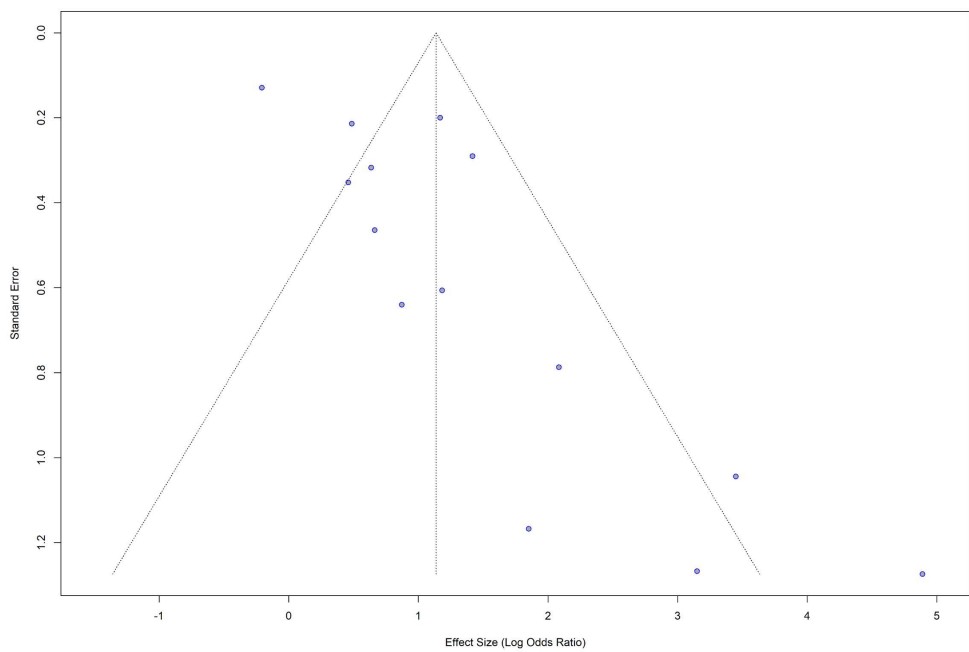

**Fig 11. Funnel plot for the studies with crude (unadjusted) ORs for hypertension as the exposure and impaired lung function as the outcome.**

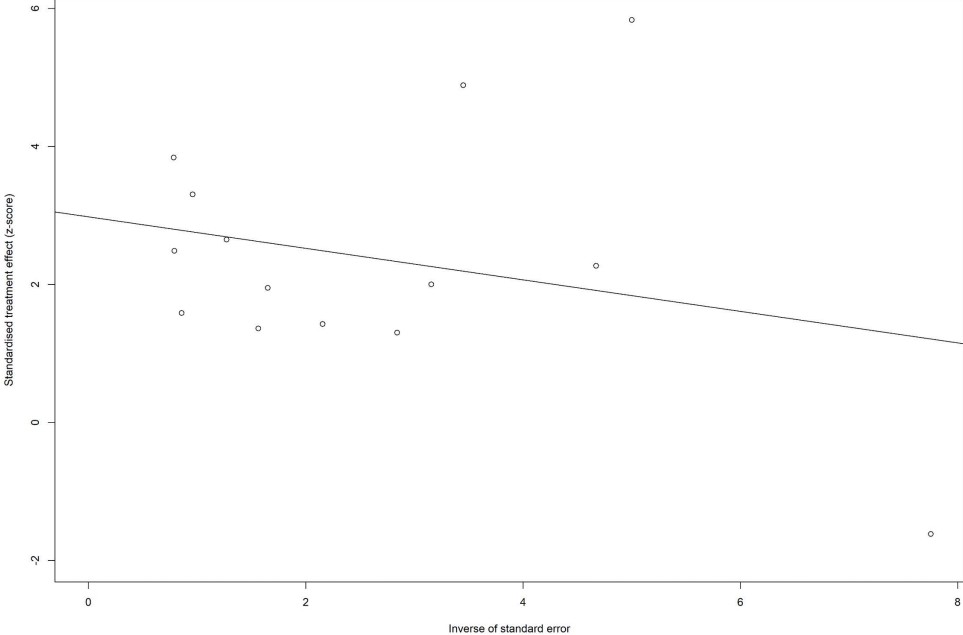

**Fig 12. Funnel plot with the regression line for the studies with crude (unadjusted) ORs for hypertension as the exposure and impaired lung function as the outcome.** The points are not symmetrically distributed around the regression line.

**Fig 13. Trim-and-fill funnel plot for the studies with crude (unadjusted) ORs for hypertension as the exposure and impaired lung function as the outcome.**

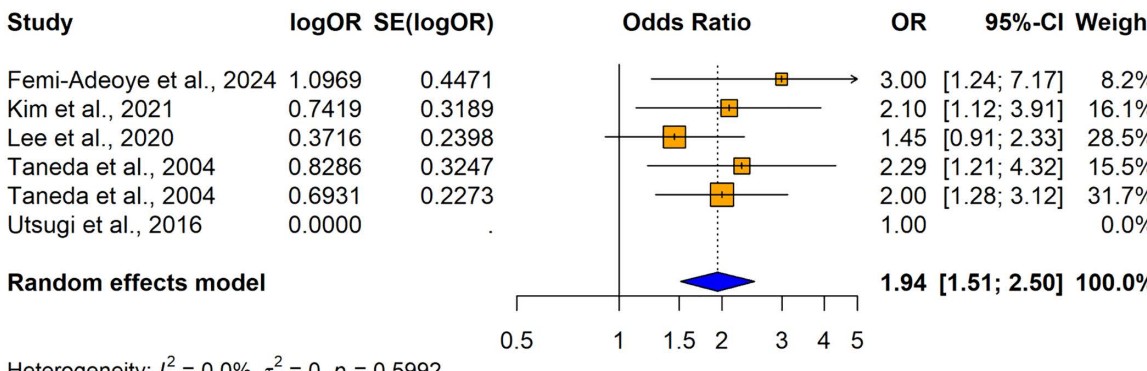

| Study | logOR | SE(logOR) | | OR | 95%-CI | Weight |
|---|---|---|---|---|---|---|
| Femi-Adeoye et al., 2024 | 1.0969 | 0.4471 | | 3.00 | [1.24; 7.17] | 8.2% |
| Kim et al., 2021 | 0.7419 | 0.3189 | | 2.10 | [1.12; 3.91] | 16.1% |
| Lee et al., 2020 | 0.3716 | 0.2398 | | 1.45 | [0.91; 2.33] | 28.5% |
| Taneda et al., 2004 | 0.8286 | 0.3247 | | 2.29 | [1.21; 4.32] | 15.5% |
| Taneda et al., 2004 | 0.6931 | 0.2273 | | 2.00 | [1.28; 3.12] | 31.7% |
| Utsugi et al., 2016 | 0.0000 | . | | 1.00 | | 0.0% |
| **Random effects model** | | | | **1.94** | **[1.51; 2.50]** | **100.0%** |

Heterogeneity: $I^2 = 0.0\%$, $\tau^2 = 0$, $p = 0.5992$

**Fig 14. Forest plot showing the association between hypertension and impaired lung function among adults when using hypertension as the exposure (using adjusted odds ratios).**

**Table 7. Summary of bidirectional associations between impaired lung function and hypertension.**

| Direction of the Association | Crude Analysis | | | | Adjusted Analysis | | | |
|---|---|---|---|---|---|---|---|---|
| | No of studies | Sample size | OR [95% CI] | $I^2$ (%) | No of studies | Sample size | OR [95% CI] | $I^2$ (%) |
| ILF ◊ HT | 31 | 192055 | 1.6957 [1.5312 – 1.8779] | 84.2 | 86 | 181371 | 1.3986 [1.3107 – 1.4925] | 59.8 |
| HT ◊ ILF | 14 | 13740 | 2.9960 [1.8637 – 4.8161] | 84.7 | 6 | 10355 | 1.9417 [1.5110 – 2.4953] | 0.0 |

**Abbreviations:** ILF: Impaired lung function; HT: Hypertension; OR: Odds Ratio; $I^2$: Residual heterogeneity/ unaccounted variability

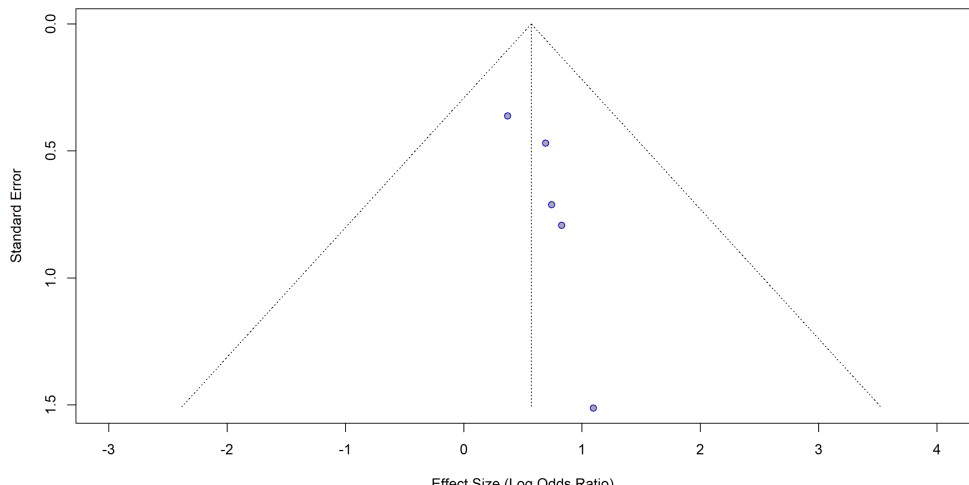

**Fig 15. Funnel plot for the studies with adjusted ORs for hypertension as the exposure and impaired lung function as the outcome.**

Three studies were imputed using the trim-and-fill method to correct the asymmetry in the funnel plot (Fig 16). After imputation the adjusted effect size was 0.4916 [95% CI: 0.0553; 0.9280], p = 0.0272. I² = 0.0% (no variability). The adjustment changed the overall conclusion indicating that there is no significant evidence of publication bias. Trim-and-fill analysis showed that three studies need to be imputed to make the results more symmetrical.

## Sensitivity analysis

Sensitivity analyses using a leave-one-out approach conducted on the meta-analyses where the impaired lung function being exposure, revealed that the results remained statistically significant even when each study was removed individually, indicating robust findings for both unadjusted and adjusted ORs (Figs 17 and 18) (S1 and S2 Tables). Furthermore,

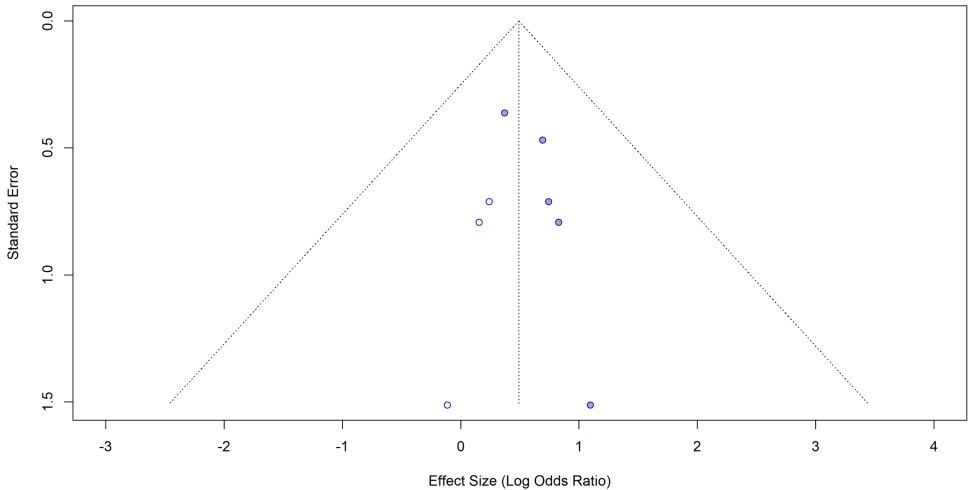

**Fig 16. Trim-and-fill funnel plot for the studies with adjusted ORs for hypertension as the exposure and impaired lung function as the outcome.**

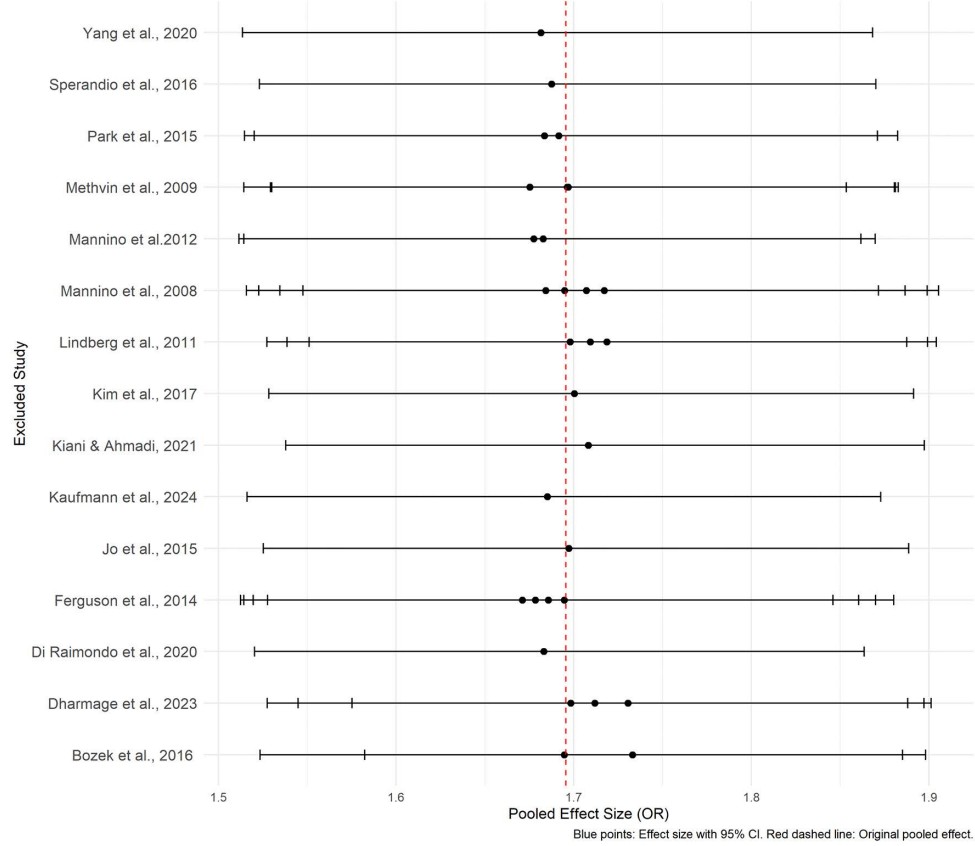

**Fig 17. The relationship between impaired lung function and hypertension. Leave-one-out sensitivity analysis of the unadjusted ORs – ILF (exposure) and HT (outcome).**

the exclusion of studies with low quality or high risk of bias did not alter the results, further supporting the robustness of the findings.

For the unadjusted ORs, sensitivity analysis performed by excluding one study at a time from the meta-analysis conducted on the effect of hypertension on impaired lung function demonstrated consistent results, with no significant changes to the overall pooled effect size (Fig 19, S3 Table). Even the substantial heterogeneity among the studies remained unchanged. These findings indicate that the overall effect size is robust and not driven by any single study, confirming the reliability of the meta-analysis results. Further the exclusion of studies with high risk of bias or low quality did not alter the heterogeneity. However, no sensitivity analysis was conducted for the adjusted OR, as the included studies showed no significant heterogeneity ($I^2 = 0.0\%$), making additional analysis unnecessary.

## Subgroup analysis

**Impaired lung function as the exposure and hypertension as the outcome.** The subgroup analyses conducted using crude ORs showed stronger pooled effects for cross-sectional studies (OR = 1.7549, 95% CI: 1.5702–1.9614) compared to cohort and case-control studies, but no significant differences were found between study designs (Q = 0.77, p = 0.6820) (Fig 20). Significant subgroup differences were observed by country (Q = 30.73, p = 0.0003), with higher odds ratios in Italy and Brazil and smaller and non-significant effects were seen in Poland and Australia. Substantial heterogeneity was observed within the United States and Korea (Fig 21). The Mannino et al. [58] study was included

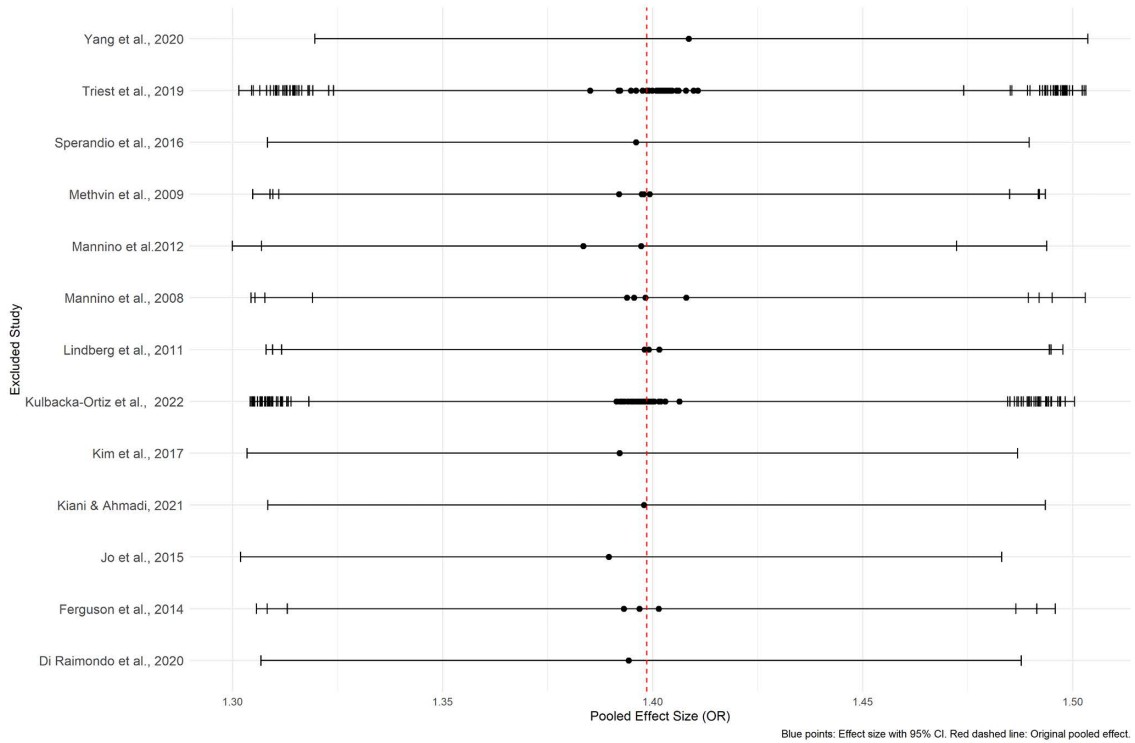

**Fig 18. The relationship between impaired lung function and hypertension. Leave-one-out sensitivity analysis of the adjusted ORs - ILF (exposure) and HT (outcome).**

by assigning it to selected countries, but the lack of country-specific sample sizes limited further disaggregation. Both obstructive (OR =1.6457; 95% CI: 1.4524–1.8648) and restrictive (OR = 1.9222 (95% CI: 1.7402–2.1231)) lung function impairments were significantly associated with the outcome, whereas the mixed lung function impairment type showed a weaker non-significant association. A significant subgroup difference was observed between the impairment types (Q = 7.32, p = 0.0257), suggesting that the strength of association varied across different patterns of lung function impairment, with restrictive type showing the strongest association (Fig 22).

The subgroup analysis conducted using adjusted ORs provided insights into variations in odds ratios across different study designs, population characteristics, and adjustments for potential confounders. Case-control studies demonstrated higher odds ratio (OR = 3.66; 95% CI: 1.2477–10.7361), though differences by study design were not statistically significant (Q = 3.09, p = 0.2128) (Fig 23). Subgroup analysis by lung function impairment type showed that restrictive impairment (OR = 1.5632, 95% CI: 1.4422; 1.6944]) had a stronger association with hypertension compared to obstructive impairment (OR = 1.2758; 95% CI: 1.1584; 1.4052), with the studies conducted to assess the restrictive impairment exhibiting lower variability ($I^2$ = 30.3%) (Fig 24). Subgroup analyses by country showed no significant differences in the pooled effects between groups (Fig 25).

Adjustments for confounders like sex, alcohol consumption levels, education, macronutrient intake (protein, carbohydrate, and total fat) and household income level significantly influenced the effect sizes (ORs), with unadjusted studies often showing higher odds ratios. Adjusting for physical inactivity and obesity results in a significantly higher odds ratio (OR = 17.51) compared to studies without such adjustments (OR = 1.40), indicating that these factors may strongly influence the observed association. Although adjustments for BMI, diabetes status, smoking, race, and dyslipidemia were observed, but differences were not statistically significant (S4 Table). The lack of statistical significance suggests that

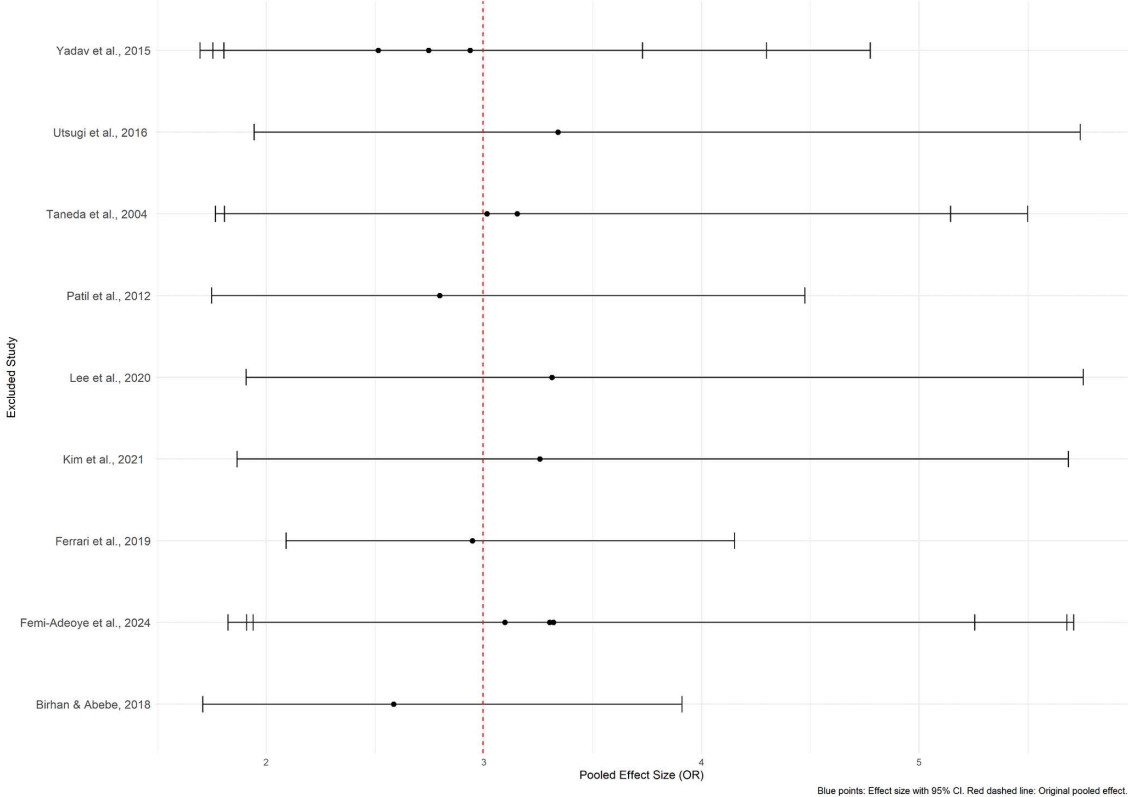

**Fig 19. The relationship between hypertension and impaired lung function. Leave-one-out sensitivity analysis of the unadjusted ORs – HT (exposure) and ILF (outcome).**

these factors do not substantially impact the overall effect in the context of this meta-analysis. A limitation of the subgroup analysis is the restricted availability of studies that included adjustments for specific variables. Of note, only one study accounted for physical inactivity, obesity, alcohol consumption, household income level, and macronutrient intake, which limits the generalizability and robustness of the findings related to these moderators.

Subgroup analysis by continent was not conducted because the Mannino et al. [58] study included data from multiple continents, and the sample sizes for individual countries within these continents were unavailable. Thus, inability to accurately assign data to specific continents, the analysis would lack validity and could introduce bias. To ensure the robustness of the findings, the continent-based subgroup analysis was omitted. Since all studies involved both male and female participants, subgroup analyses by gender were not performed.

### Hypertension as the exposure and Impaired lung function as the outcome

The subgroup analyses conducted using crude ORs showed no statistically significant differences in odds ratios (ORs), between cross-sectional [OR = 2.4158, 95% CI: 1.8544–3.1472, $I^2$ = 55.7%, p = 0.00206] and case-control studies [OR = 8.1920, 1.4427–46.5157, $I^2$ =87.9%, p < 0.0001] despite substantial variability in effect sizes, indicating study design did not significantly influence the association (Q = 1.86, p = 0.1731) (Fig 26). In contrast, significant variability was observed by country (Q = 74.48, p < 0.0001) (Fig 27) and continent (Q = 56.41, p < 0.0001) (Fig 28), suggesting the association between hypertension and impaired lung function is influenced by geographic factors, with moderate heterogeneity within Asia ($I^2$ = 69.6%). Notable country-level variations in effect sizes were identified, with the highest ORs reported in studies from

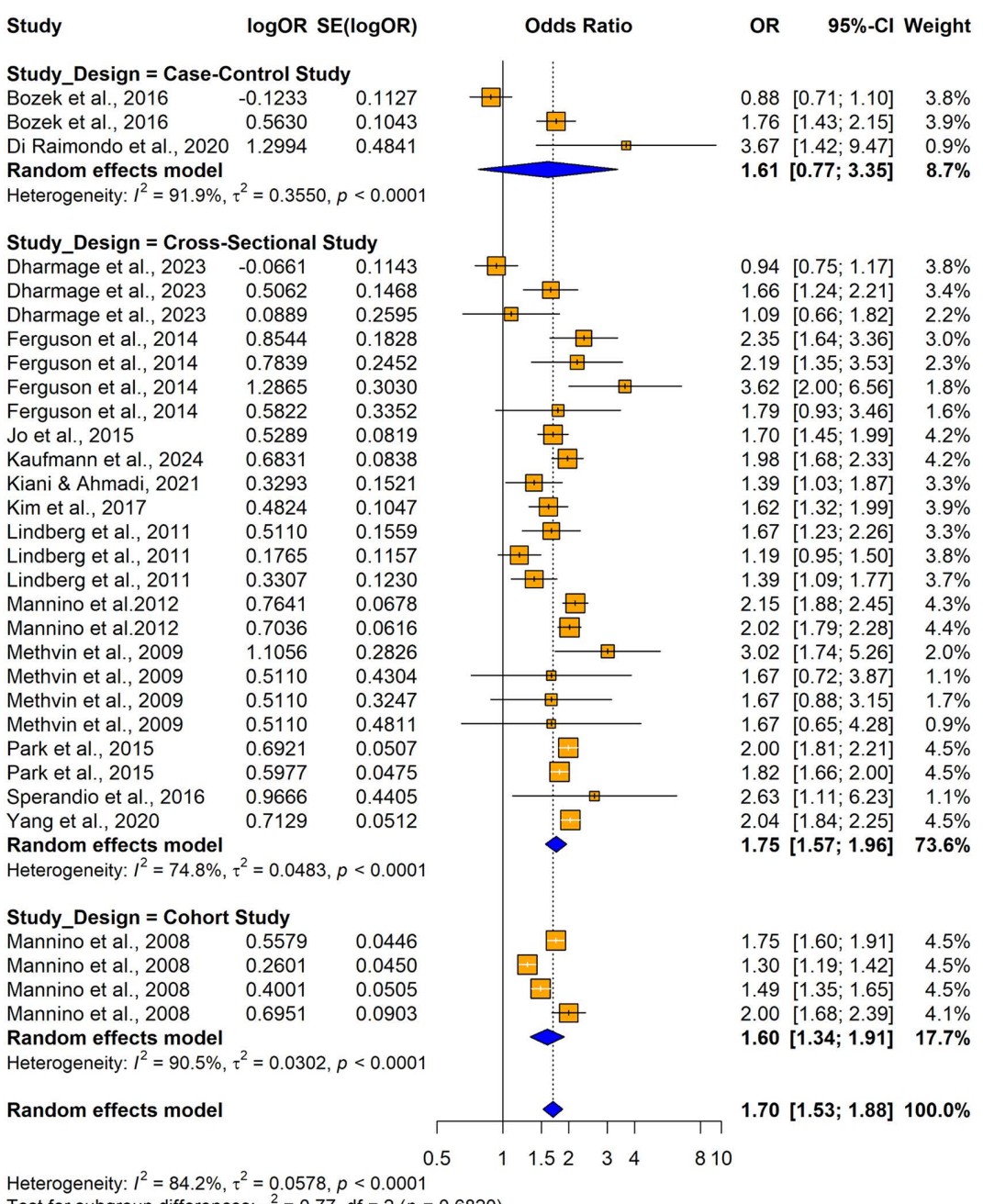

**Fig 20. Forest plot for subgroup analyses by study design (unadjusted ORs): impaired lung function as the exposure and hypertension as the outcome.**

Ethiopia OR (31.50; 95% CI: 4.0730–243.619] and India (OR = 16.2955; 95% CI: 4.7113–56.3634) and the lowest in studies from Japan (OR = 1.6250; 95% CI: 1.0672–2.4744) and Italy (OR = 0.8120; 95% CI: 0.6310–1.0450), emphasizing the importance of regional factors. Analysis by lung function impairment type revealed the strongest association for mixed impairment (OR = 23.3330; 95% CI [1.9481–279.4615], followed by restrictive (OR = 6.3229; 95% CI [1.4644–27.3007]),

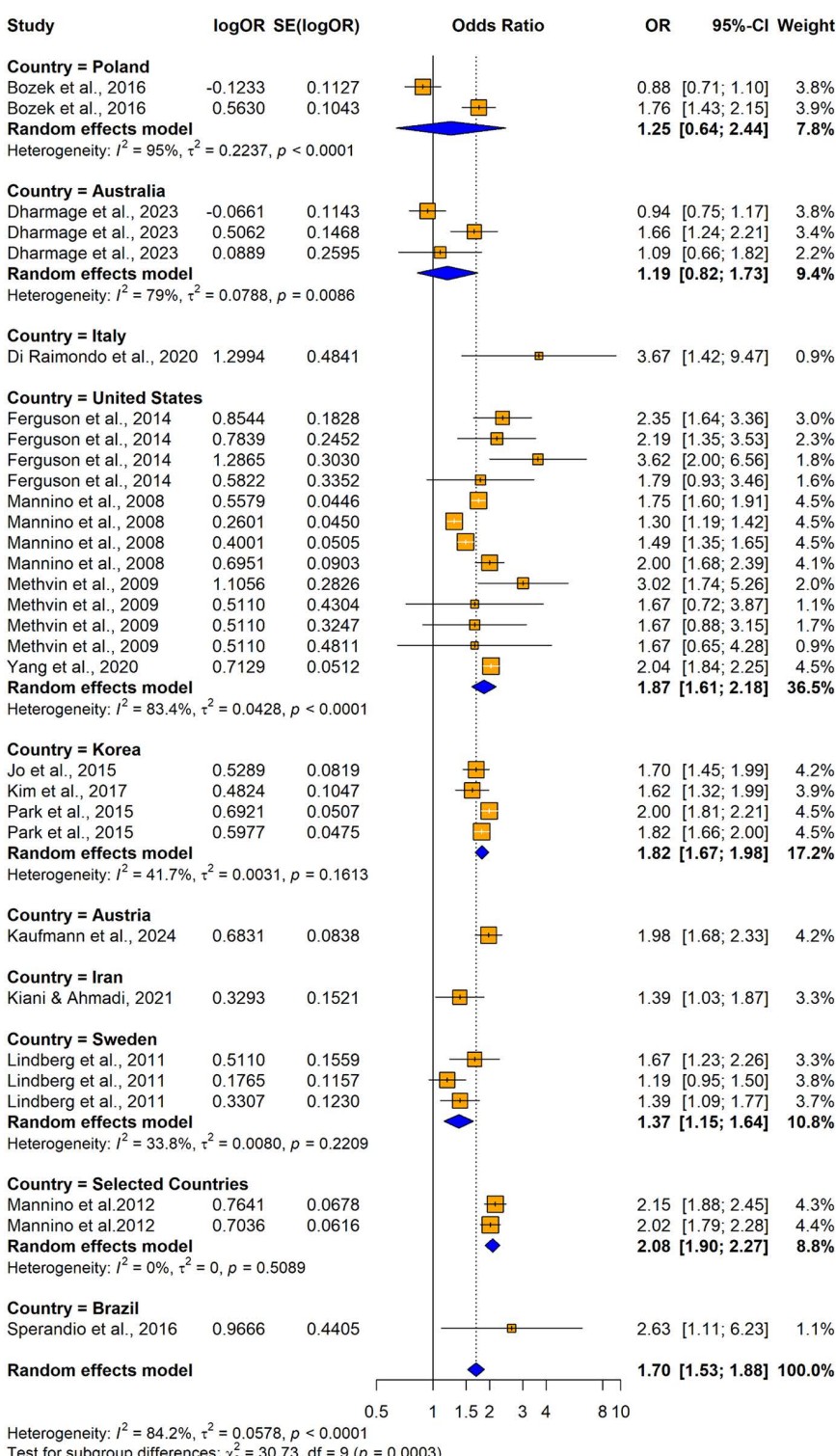

**Fig 21. Forest plot for subgroup analyses by country (unadjusted ORs): impaired lung function as the exposure and hypertension as the outcome.**

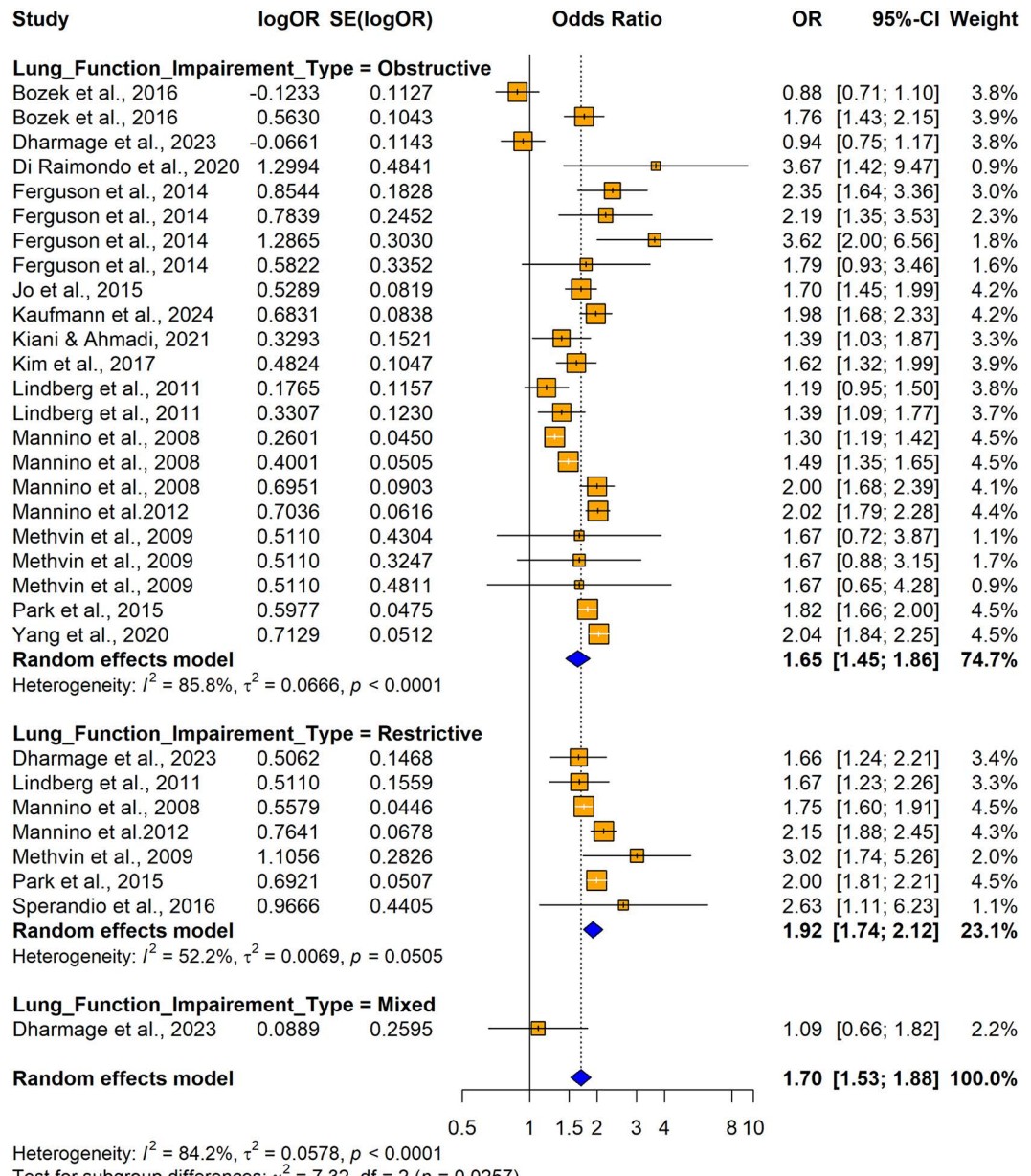

**Fig 22. Forest plot for subgroup analyses by lung function impairment type (unadjusted ORs): impaired lung function as the exposure and hypertension as the outcome.**

obstructive impairments (OR = 2.2477; 95% CI [1.2944–3.9032]), and restrictive and obstructive impairment (OR = 1.8860; 95% CI [1.0137–3.5089]) though differences between subgroups were not statistically significant (Q = 5.54, p = 0.1362) (Fig 29). As all studies involved both male and female participants, subgroup analyses by gender were not performed.

The subgroup analysis conducted using adjusted ORs indicated that there were no statistically significant differences in ORs based on countries, continent, or adjustments for variables such as sex, education, physical inactivity, alcohol

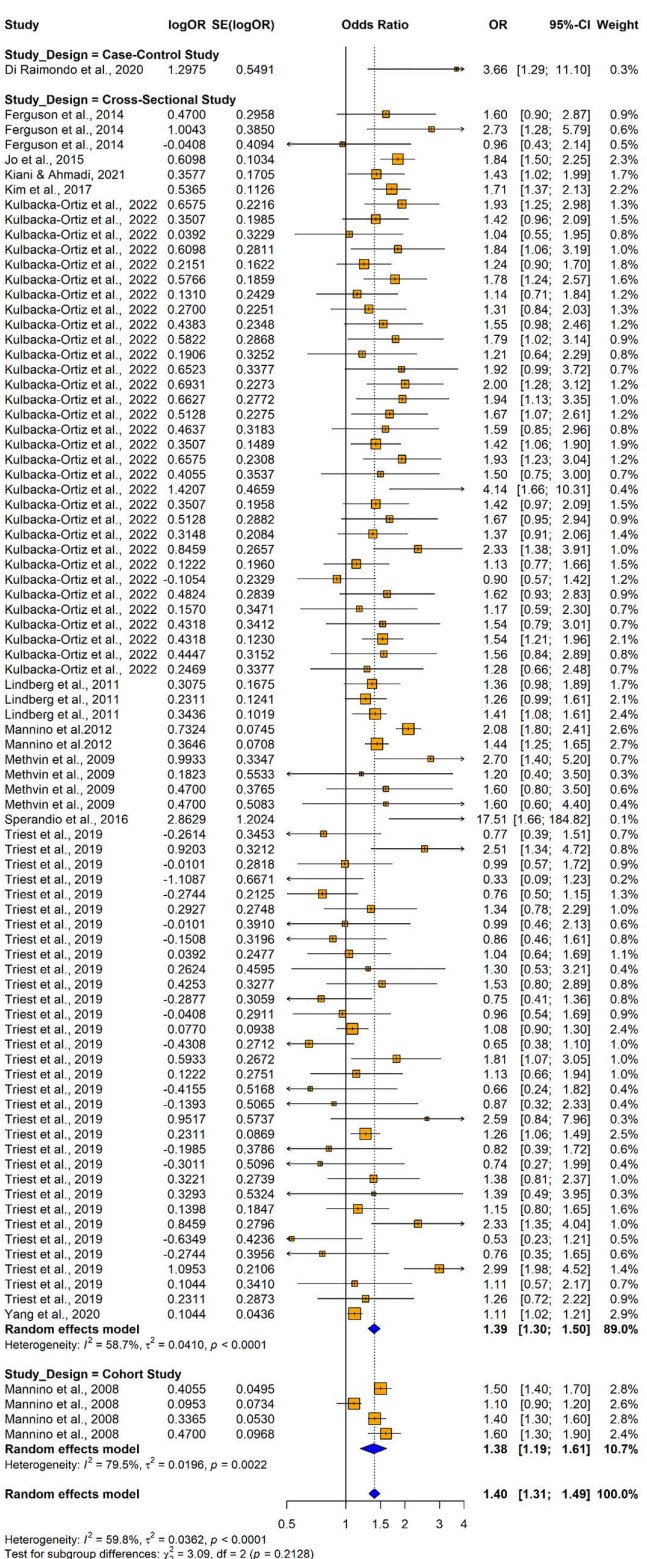

**Fig 23. Forest plot for subgroup analyses by study design (adjusted ORs): impaired lung function as the exposure and hypertension as the outcome.**

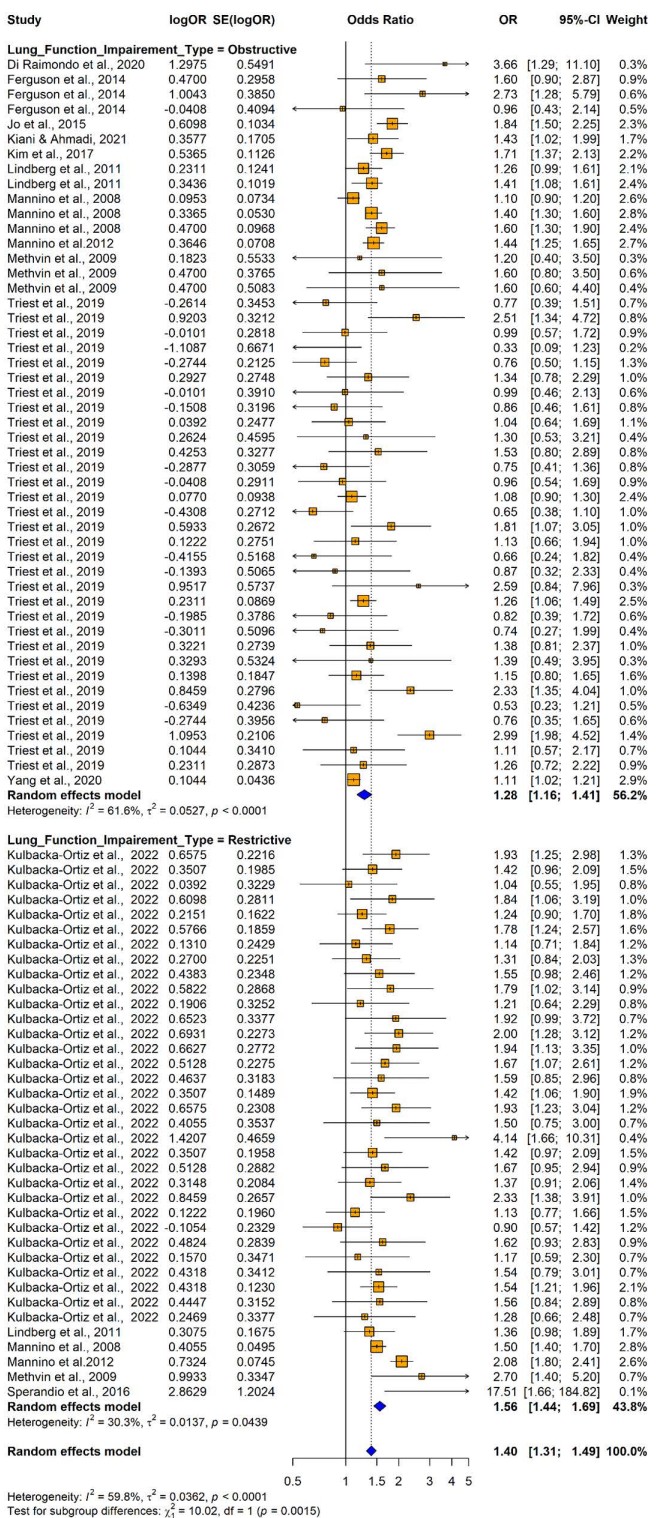

**Fig 24. Forest plot for subgroup analyses by lung function impairment type (adjusted ORs): impaired lung function as the exposure and hypertension as the outcome.**

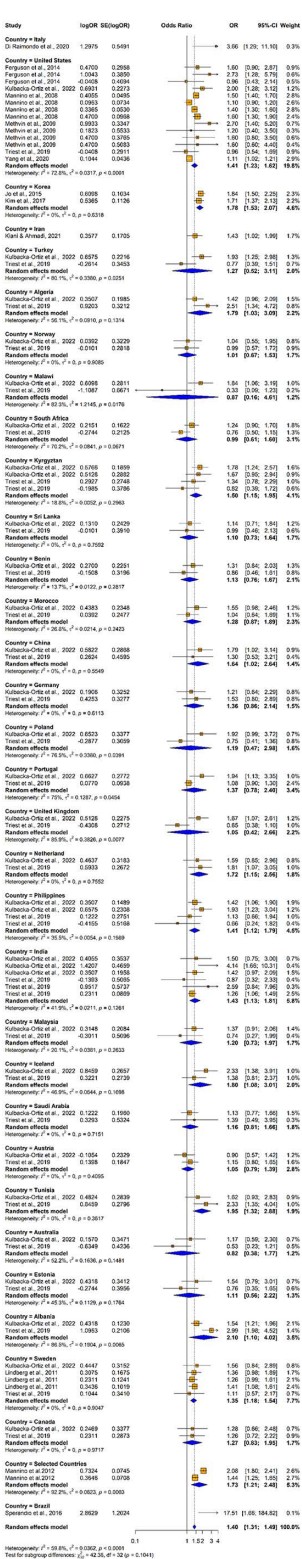

**Fig 25. Forest plot for subgroup analyses by country (adjusted ORs): impaired lung function as the exposure and hypertension as the outcome.**

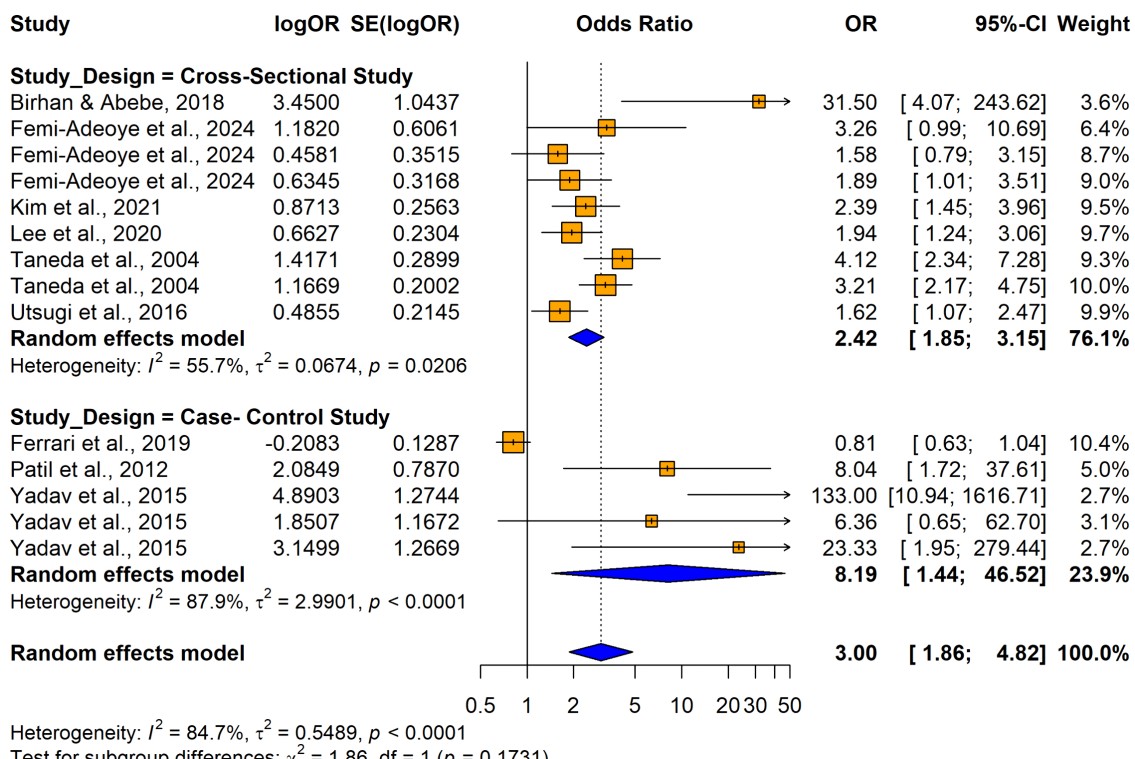

Fig 26. Forest plot for subgroup analyses by study design (unadjusted ORs): hypertension as the exposure.

consumption, smoking, residence, household income level diabetes status, or waist circumference (p > 0.05) (Figs 30 and 31) (S5 Table). The analysis showed slight variations in odds ratios, with a stronger association for obstructive lung impairment (OR = 2.0331; 95% CI: 1.4145–2.9223) compared to restrictive impairments (OR = 1.7292; 95% CI: 1.1182–2.6740), but this difference was not statistically significant (Q = 1.25, p = 0.5358) (Fig 32).

The study from Japan [39] reports an odds ratio (OR) of 1.00, indicating no effect, and was excluded from the sub group analysis due to the lack of variation in its result. All included studies were cross-sectional; therefore, no subgroup analysis was performed by the study design due to a lack of variation in the study design. Subgroup analyses could not be performed for variables such as age, gender, BMI, and other comorbidities because either most studies failed to report primary data on these associations, or all the included studies were adjusted for similar factors including age, BMI, dyslipidemia, race (black or white), obesity, CRP and macronutrient intake (protein, carbohydrate, and total fat), leaving no variability to form distinct subgroups.

## Meta-regression

In the univariate meta-regression of impaired lung function being the exposure, adjusting for certain moderators such as age, waist circumference, residence and CRP were flagged as redundant variables and dropped from the models due to collinearity or overlap with other predictors. This indicates that their effects could not be uniquely distinguished in the analysis. Adjusting for sex (and education (were negatively associated with the effect size, suggesting that including these adjusted factors in the model reduces the observed effect. Despite the inclusion on moderators, heterogeneity remained moderate (Table 8).

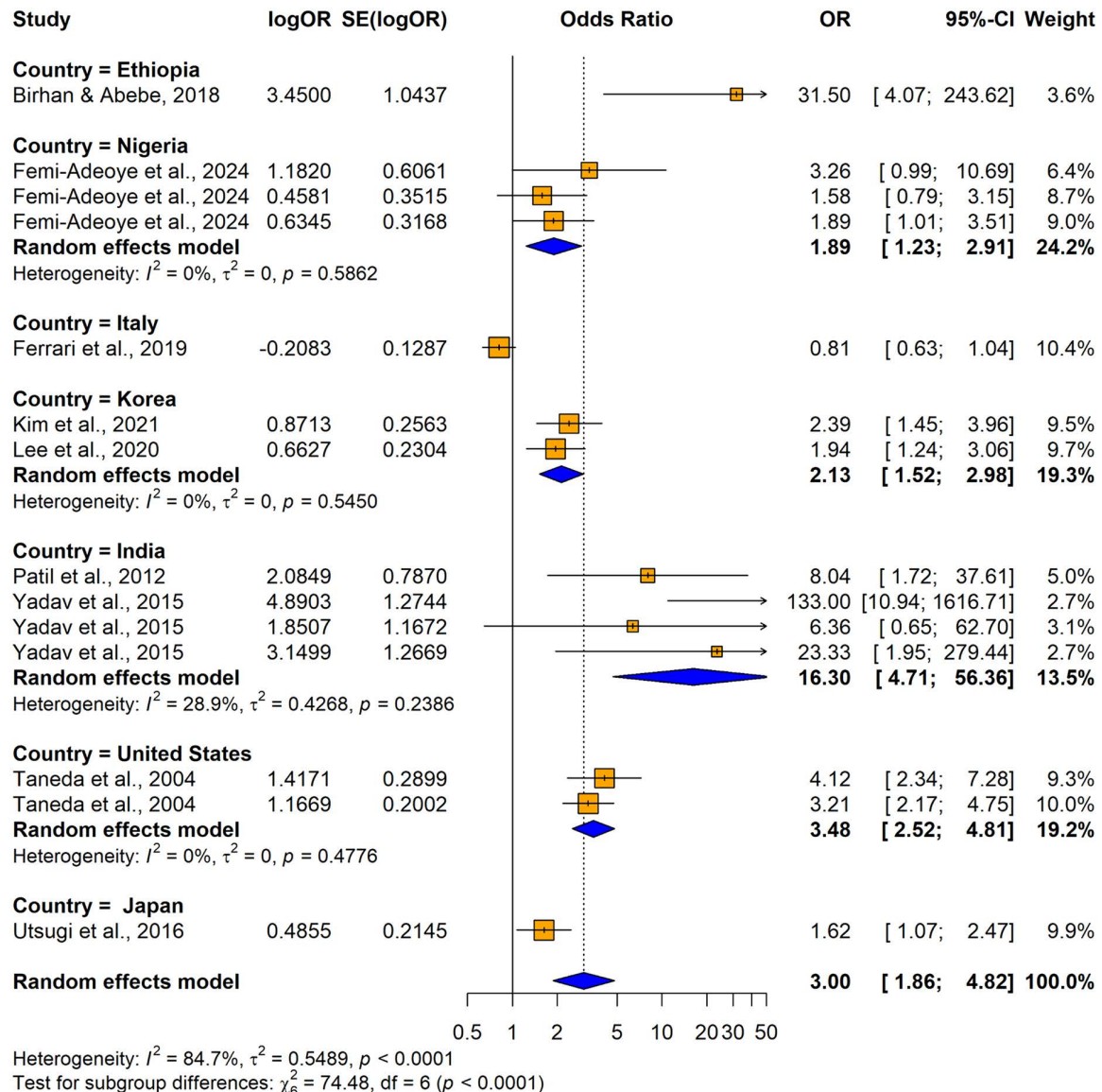

**Fig 27. Forest plot for subgroup analyses by country (unadjusted ORs): hypertension as the exposure.**

An initial meta-regression model including all potential moderators (both significant and non-significant from univariate analysis), resulted in high residual heterogeneity ($I^2$=99.12%), suggesting that the inclusion of non-significant variables did not improve the explanatory power of the model ($R^2$=3.44%). During the multivariable meta-regression which included only the significant variables from the univariate meta-regression (sex and education), the model explained 32.56% of the heterogeneity in effect sizes. However, neither of the adjusted moderators was statistically significant (Table 9), suggesting that these variables did not significantly contribute to the variation in impaired lung function among individuals with hypertension in this analysis.

## Discussion

To the best of our knowledge, this is the first systematic literature review and meta-analysis to date comprehensively evaluating the association between impaired lung function and hypertension and vice versa using data drawn from

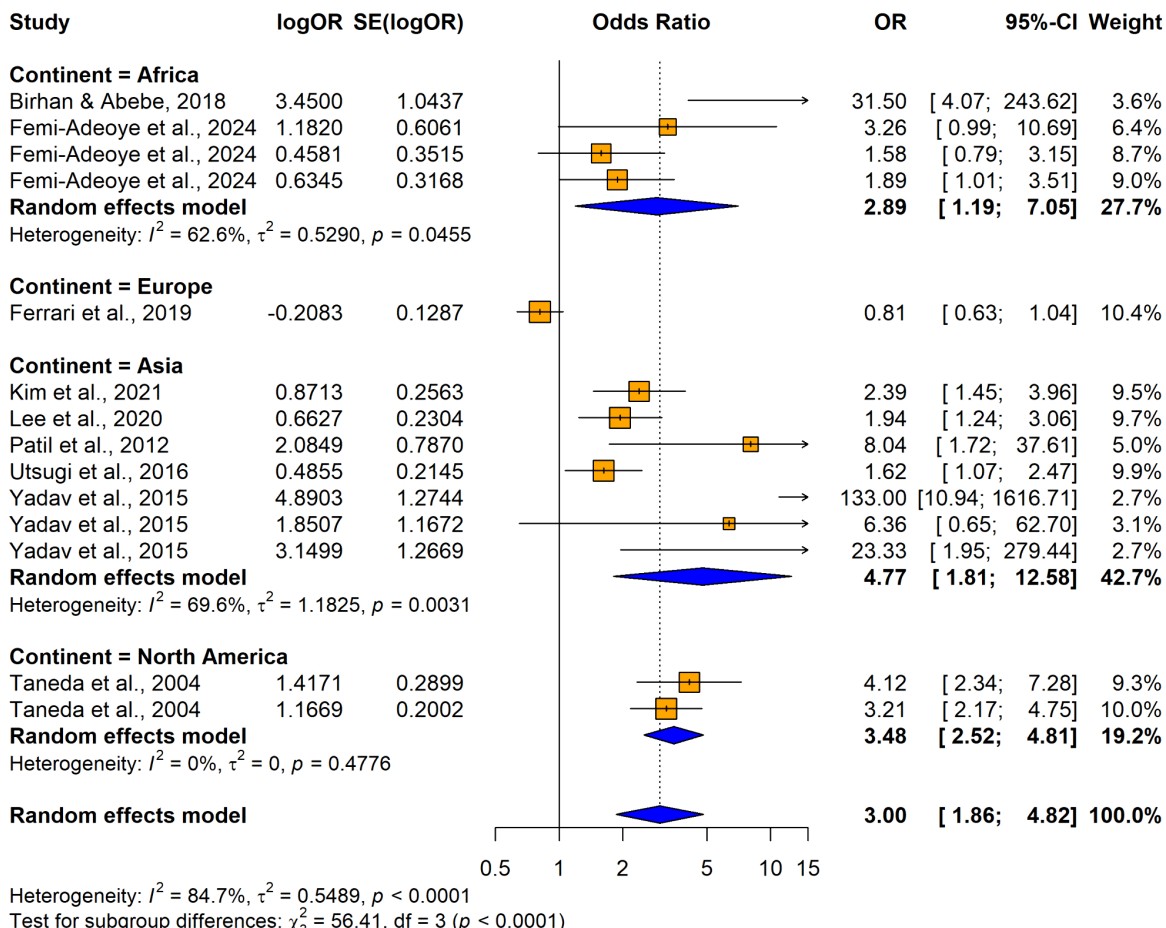

**Fig 28. Forest plot for subgroup analyses by continent (unadjusted ORs): hypertension as the exposure.**

observational studies conducted globally. Our findings revealed that the pooled unadjusted effect of lung function impairment on hypertension was 1.70 (95% CI: [1.53–1.88]) with substantial heterogeneity (I²=84.2%, p<0.0001), while the pooled adjusted effect of lung impairment on hypertension was lower at 1.40 (95% CI: [1.31–1.49]), with moderate heterogeneity (I²=59.8%, p<0.0001). This suggests that adjustment for confounding factors reduced the strength of the association between hypertension and lung function impairment (Table 7). In contrast, the pooled unadjusted effect of hypertension on lung impairment was 3.00 (95% CI: [1.86–4.82]) with considerable heterogeneity (I²=84.7%, p<0.0001), but the pooled adjusted effect decreased to 1.94 (95% CI: [1.51–2.50]), with no significant heterogeneity (I²=0.0%, p=0.5992). This indicates that adjustments for potential confounders significantly attenuated the association between hypertension and lung impairment, suggesting that the initial unadjusted effect might have been overestimated due to confounding factors. Thus, in the unadjusted analysis, hypertension had a stronger effect (OR = 3.00) when used as the exposure, compared to lung function impairment's effect on hypertension (OR = 1.70). However, after adjustments, both associations exhibit a decline, with hypertension still showing a greater effect on lung impairment (OR = 1.94) compared to effect of lung function impairment on hypertension (OR = 1.40). Although adjusted effect sizes were numerically larger when hypertension was considered as the exposure, differences in study design, number of included studies, and publication bias limit direct comparison of magnitude between the two directions.

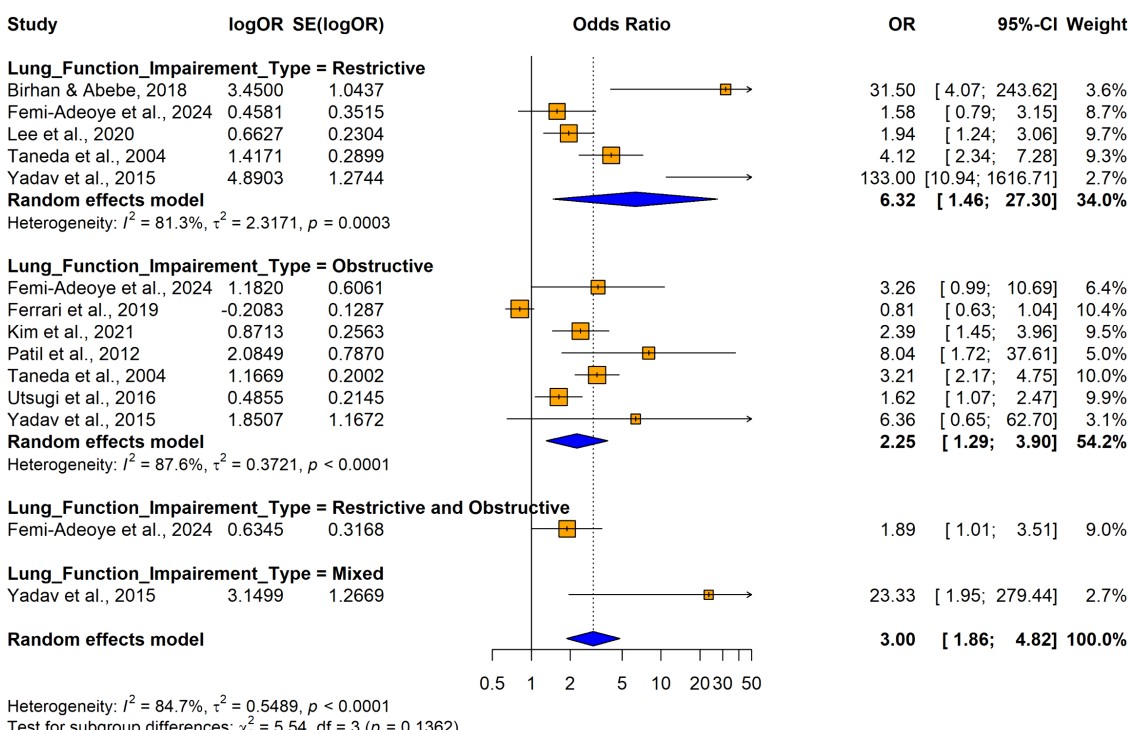

**Fig 29. Forest plot for subgroup analyses by lung function impairment type (unadjusted ORs): hypertension as the exposure.**

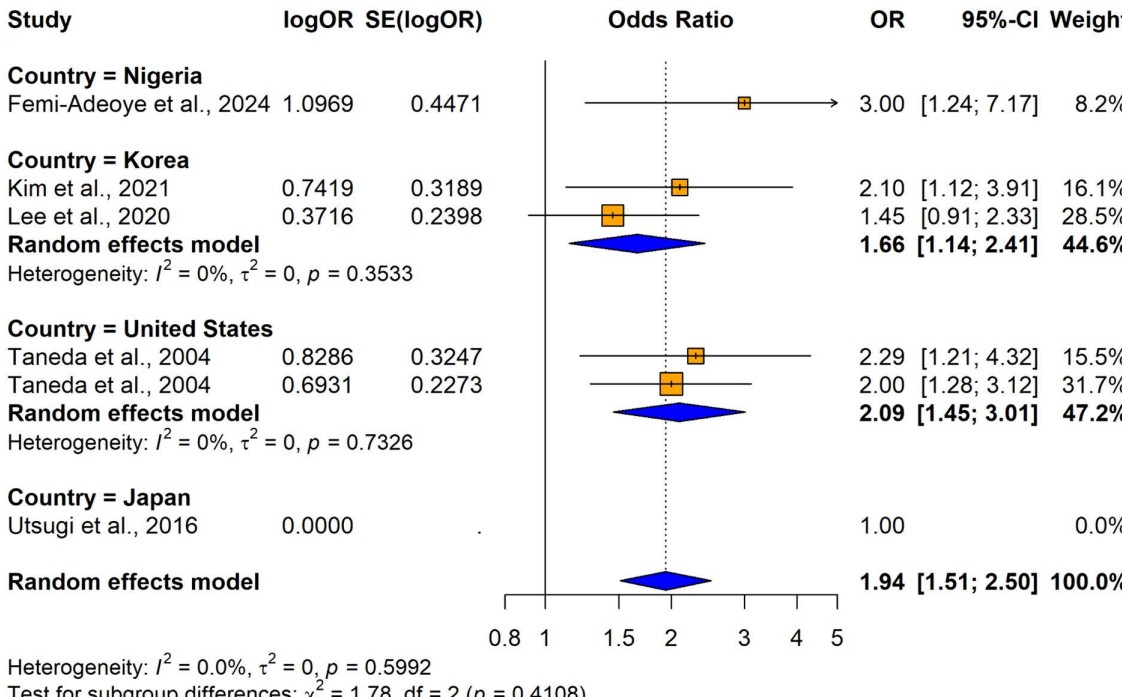

**Fig 30. Forest plot for subgroup analyses by country (adjusted ORs): hypertension as the exposure.**

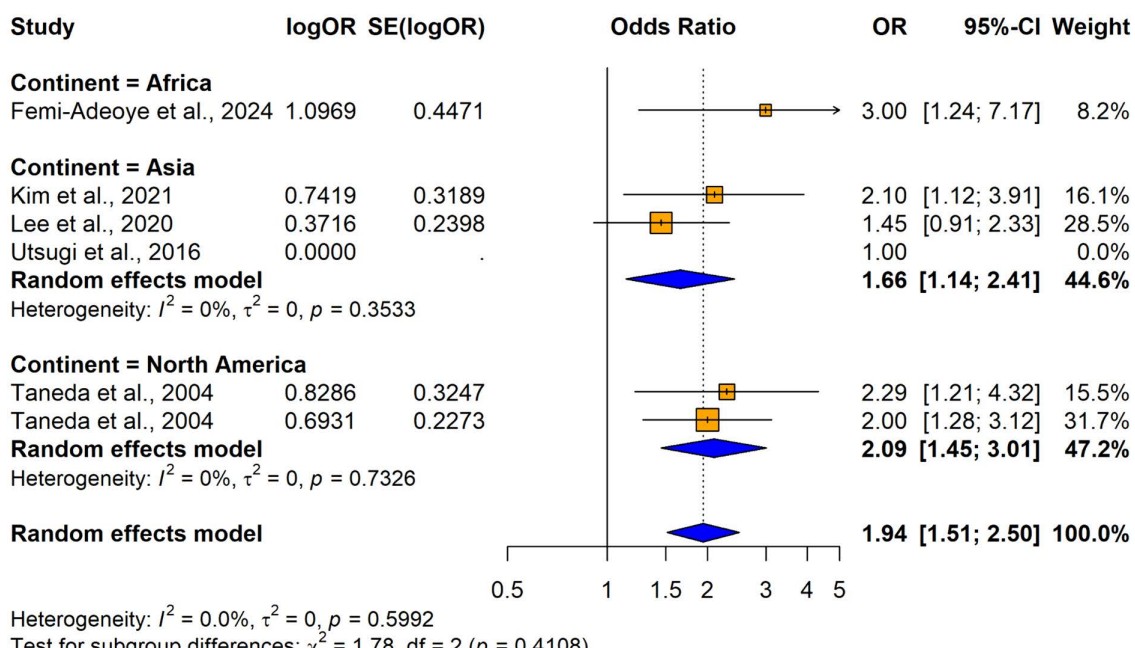

**Fig 31. Forest plot for subgroup analyses by continent (adjusted ORs): hypertension as the exposure.**

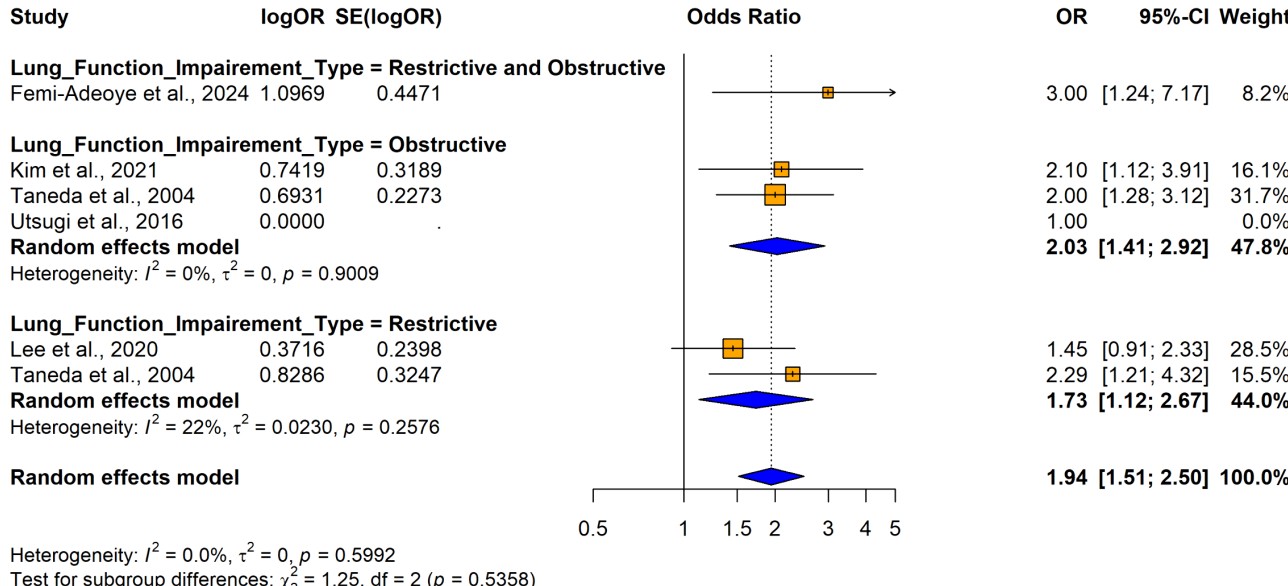

**Fig 32. Forest plot for subgroup analyses by lung function impairment type (adjusted ORs): hypertension as the exposure.**

Although association of HTN with altered pulmonary function is not a new observation [26,66,67], the exact mechanisms underlying the bi-directional relationship between hypertension and impaired lung function remain obscure. Several studies have demonstrated that impaired lung function is associated with elevated levels of inflammatory markers [68–72],

**Table 8. Results of the univariate meta-regression with impaired lung function as the exposure.**

| Moderator | Estimate | SE | Z value | p value | 95% CI | $I^2$ (%) | $R^2$ (%) | $\tau^2$ |
|---|---|---|---|---|---|---|---|---|
| BMI | −0.4153 | 0.2862 | −1.4507 | 0.1469 | −0.9763 to 0.1458 | 49.31 | 4.77 | 0.0438 (SE = 0.0176) |
| Sex | −0.3780 | 0.1249 | −3.0270 | 0.0025 | −0.6227 to 0.1332 | 40.39 | 32.04 | 0.0312 (SE = 0.0145) |
| Smoking | −0.4241 | 0.2844 | −1.4910 | 0.1360 | −0.9816 to 0.1334 | 49.30 | 4.83 | 0.0437 (SE = 0.0176) |
| Race | 0.0791 | 0.1146 | 0.6901 | 0.4901 | −0.1455 to 0.3036 | 49.51 | 0.00 | 0.0476 (SE = 0.0188) |
| Physical Inactivity | 2.6597 | 46.7245 | 0.0569 | 0.9546 | −88.9186 to 94.2381 | 50.92 | 0.00 | 0.0460 (SE = 0.0180) |
| Education | −0.2768 | 0.0978 | −2.8287 | 0.0047 | −0.4685 to −0.0850 | 41.17 | 29.84 | 0.0323 (SE = 0.0147) |
| Alcohol Consumption Levels | −0.1025 | 0.2297 | −0.4461 | 0.6555 | −0.5527 to 0.3478 | 47.10 | 0.00 | 0.0487 (SE = 0.0192) |
| Diabetes Status | 0.0709 | 0.1841 | 0.3849 | 0.7003 | −0.2900 to 0.4317 | 47.63 | 0.00 | 0.0490 (SE = 0.0192) |
| Dyslipidemia | −0.1024 | 0.2297 | −0.4458 | 0.6557 | −0.5526 to 0.3478 | 47.10 | 0.00 | 0.0487 (SE = 0.0192) |
| Household Income Level | −0.1025 | 0.2297 | −0.4461 | 0.6555 | −0.5527 to 0.3478 | 47.10 | 0.00 | 0.0487 (SE = 0.0192) |
| Obesity | 2.6597 | 46.7245 | 0.0569 | 0.9546 | −88.9186 to 94.2381 | 50.92 | 0.00 | 0.0460 (SE = 0.0180) |
| Macronutrient Intake | −0.1025 | 0.2297 | −0.4461 | 0.6555 | −0.5527 to 0.3478 | 47.10 | 0.00 | 0.0487 (SE = 0.0192) |

**Note:** Variables flagged as redundant due to collinearity (age, waist circumference, residence, and CRP) were excluded from the univariate meta-regression analyses. $I^2$: residual heterogeneity/ unaccounted variability; $R^2$: amount of heterogeneity accounted for, $\tau^2$: estimated amount of residual heterogeneity.

**Table 9. Results of the multivariable meta-regression: significant moderators from univariate analysis (exposure: impaired lung function).**

| Moderator | Estimate | SE | Z value | p value | 95% CI |
|---|---|---|---|---|---|
| Sex | −0.2520 | 0.1759 | −1.4326 | 0.1520 | −0.5967 to 0.0928 |
| Education | −0.1386 | 0.1367 | −1.0141 | 0.3105 | −0.4066 to 0.1293 |

such as circulating C-reactive protein (CRP), and interleukin 6 causing increased risk of hypertension [69,70]. The pathophysiology of elevated cardiovascular risk in patients with chronic lung disease is multifactorial [73]. The overactivation of the sympathetic nervous system due to systemic inflammation [74–76], hypoxia [77], reduced baroreceptor sensitivity [78], inhibition of pulmonary stretch receptors [79], elevated muscle sympathetic nerve activity [80], inhibition of parasympathetic nervous system [73] together with vascular stiffness [78] result in hypertension with or without atherosclerosis (Fig 33). Further, higher forced vital capacity (FVC) is supposed to be a negative predictor of developing hypertension [25,26].

Hypertension is linked to both increased systemic and pulmonary vascular resistance and increased vascular stiffness. Given the highly vascular nature of the lung and intimate anatomic coupling of vascular and parenchymal elements, several theories have been proposed to explain the relationship between hypertension and impaired lung function [81]. One proposed mechanism is that hypertension leads to left ventricular dysfunction, subsequently increasing left atrial pressure leading to elevated pulmonary artery pressure, which contributes to interstitial lung edema, and reduces lung compliance and functional residual capacity, ultimately lowering $FEV_1$ and FVC values [82]. In addition to that, the increased peripheral vascular resistance associated with hypertension may impact the structure and function of the lungs and pulmonary circulation, causing small airway damage and impaired lung function [83].

The pooled odds ratio remained significant in several subgroup analyses when impaired lung function was considered as the exposure. However, many subgroup categories included only a single study, limiting the ability to draw firm conclusions and requiring cautious interpretation of these findings.

The findings indicate obstructive, restrictive and mixed lung function impairments are significantly associated with hypertension, with restrictive lung impairment consistently showing a stronger association with hypertension compared to obstructive impairment when impairment lung function is the exposure. Although subgroup differences were not

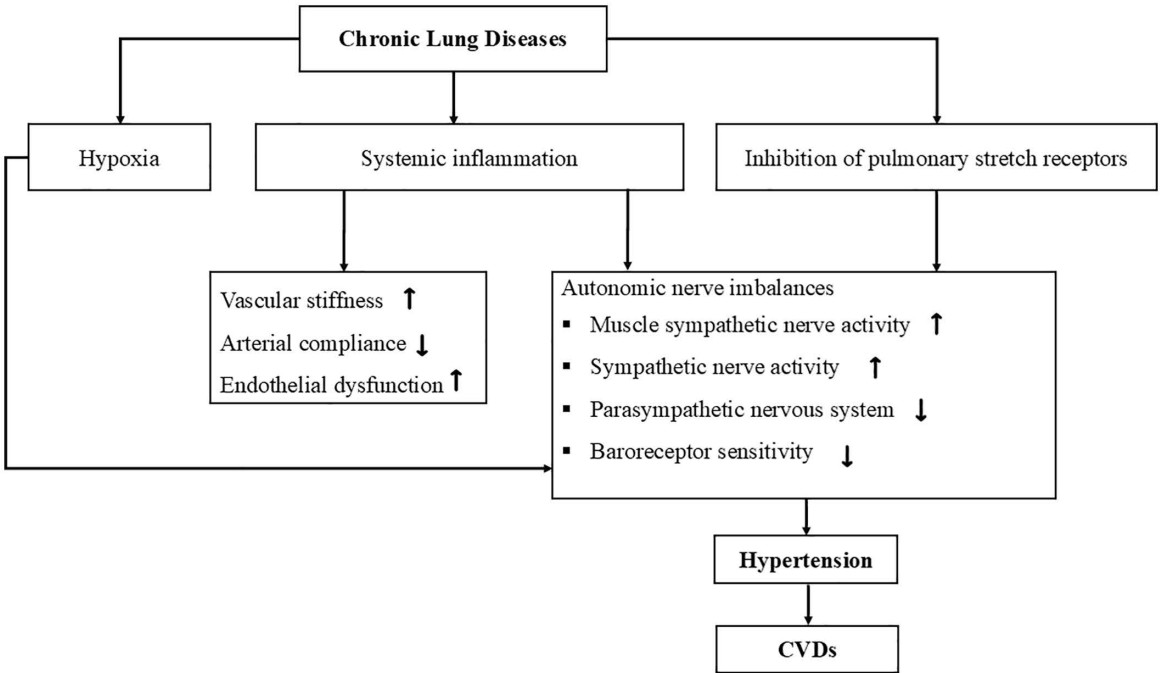

**Fig 33. Pathophysiology of hypertension leading to CVD in patients with impaired lung function [73].**

statistically significant hypertension as the exposure shows a stronger association with mixed lung impairment. Adjustments reduce the strength of associations, highlighting the potential influence of confounding factors. These findings suggest that restrictive impairment may have a more consistent and robust association with hypertension, while mixed and obstructive impairments show stronger associations in certain contexts. This highlights the importance of accounting for lung impairment types and study design variability when examining the bidirectional relationship between hypertension and lung function.

Importantly, the stronger and more consistent association observed between restrictive lung impairment and hypertension may reflect shared cardiometabolic and mechanical pathways. Restrictive ventilatory patterns are frequently linked to obesity, and are characterized by reduction in lung volumes, including forced vital capacity and total lung capacity. These changes are largely driven by mechanical limitations imposed by excess adipose tissue in the thoracic-abdominal region, which restrict diaphragmatic movement and reduce chest wall compliance, thereby contributing to restrictive ventilatory physiology [84]. Beyond mechanical restriction, obesity is increasingly recognized as a chronic systemic inflammatory state. Dysfunctional adipocytes release pro-inflammatory adipokines such as interleukin-6 and leptin and cytokines that promote insulin resistance, endothelial disfunction, and vascular stiffness, which are the key pathophysiological processes underlying hypertension [85,86]. These inflammatory mediators may also adversely affect pulmonary microvascular integrity and lung tissue elasticity, leading to reduced lung compliance and restrictive physiological changes [85,87,88]. Additionally, central adiposity plays a particularly important role in respiratory impairment by increasing intra-abdominal and intrathoracic pressures, limiting lung expansion, and reducing functional residual capacity. These mechanical alterations increased respiratory workload and reduced respiratory muscle efficiency, further contributing to restrictive lung dysfunction [88]. Metabolic syndrome, which frequently coexists with obesity, provides an additional mechanistic link between restrictive lung impairment and hypertension. It is characterized by insulin resistance, chronic low-grade inflammation, and neurohormonal activation, all of which contribute to vascular stiffness and cardiometabolic dysregulation [89]. These

systemic alterations may simultaneously affect vascular and pulmonary structures, reinforcing the observed association between restrictive lung impairment and hypertension. Therefore, these findings suggest that restrictive lung impairment may serve as a clinical marker of systemic cardiometabolic dysfunction rather than isolated pulmonary pathology. This may explain why restrictive patterns demonstrate a more consistent association with hypertension compared with obstructive impairment, which is more closely related to airway-specific inflammatory processes. Importantly, this finding constitutes one of the key contributions of the present study, offering novel insight into the cardiometabolic significance of restrictive lung impairment in relation to hypertension.

In cross-sectional studies, a significant association was observed in both unadjusted and adjusted analyses when impaired lung function (ILF) was the exposure, as well as when hypertension was the exposure. Subgroup analysis indicated that study design did not significantly influence the associations, as no significant differences were found between study designs in both unadjusted and adjusted analyses for either exposure-outcome relationship (Q values: 0.77–3.09, p > 0.05). Since cross-sectional studies do not support temporal associations, the findings should be interpreted cautiously. Limited longitudinal evidence from a single cohort study suggests that impaired lung function may precede hypertension; however, comparable longitudinal evidence is lacking to support temporality in the reverse direction. Therefore, these results support significant associations in both directions, but do not establish causality except limited evidence when impaired lung function is considered the exposure.

Although sex and education were significantly associated with effect size in univariate meta-regression, these associations were no longer statistically significant in the multivariable model despite explaining a substantial proportion of heterogeneity. The loss of statistical significance in the multivariable meta-regression likely reflects collinearity among study-level characteristics such as sex and education, reduced statistical power due to the modest number of included studies relative to the number of moderators examined, and limited variability of several moderators across studies. These factors may have reduced the ability of the model to detect independent moderator effects despite their apparent influence in univariate analyses.

The quality assessment of the included studies provided crucial insight into the strength and reliability of the evidence. When impaired lung function was considered as the exposure, 17.65% of studies were classified as high quality, suggesting a limited but robust subset of evidence in this direction. The majority of studies (70.6%) were of average quality, indicating a reasonable level of methodological soundness but with potential limitations, such as incomplete adjustments for confounders or moderate heterogeneity. Only 11.76% of studies were categorized as low quality, suggesting minimal risk of severely biased evidence in this analysis. Conversely, when impaired lung function was considered the outcome, the distribution shifted slightly, with 81.8% of studies classified as high or average quality, indicating a balanced level of reliability in this direction. However, 18.2% of studies were of low quality, raising concerns about potential biases in these findings. Such variability emphasizes the importance of interpreting the findings with caution, particularly where lower-quality studies contribute to the pooled estimates. The results highlight a need for future studies with rigorous designs and comprehensive adjustments to better clarify the bidirectional relationship between hypertension and impaired lung function.

Furthermore, publication bias was evident in some analyses, as indicated by asymmetry in the funnel plots although the Egger's test results did not indicate any significance. Adjustments using trim-and-fill method shifted the effect sizes (adjusted ORs) strengthening the robustness of the results when hypertension was considered as the exposure, suggesting that publication bias or small-study effects may have influenced the initial estimates. The exclusion of studies with low quality or high risk of bias did not alter the results, further supporting the robustness of the findings.

Although, most studies defined hypertension using the widely accepted clinical threshold of systolic blood pressure ≥ 140 mmHg and/or diastolic blood pressure ≥ 90 mmHg, some incorporated antihypertensive medication use or self-reported physician diagnosis to identify treated or previously diagnosed hypertensives, which may introduce differences in definition. Similarly, an important source of methodological variability across studies relates to differences in

the definition of impaired lung function. While some studies defined impairment using the fixed ratio threshold (FEV$_1$/FVC<0.70), others applied the lower limit of normal (LLN,<5$^{th}$ percentile) of the FEV$_1$/FVC ratio based on age, sex, and population specific reference equations. These differing definitions can lead to systematic differences in case classification across studies. The fixed ratio approach does not account for the physiological decline in lung function with aging and may therefore overestimate impairment in older populations while underestimating it in younger individuals, resulting in potential age-related misclassification bias. In contrast, LLN provides individualized thresholds and reduces such misclassification by accounting for demographic variability [90]. Differences were also observed in the way airflow obstruction was classified using GOLD versus study-specific spirometric cutoff values. Because GOLD staging reflects standardized classification categories, whereas study-specific definitions may capture broader patterns of lung function impairment, these differences may affect comparability across studies and contribute to heterogeneity in pooled estimates.

## Strengths and limitations

This systematic review and meta-analysis has several strengths. To our knowledge, this is the first comprehensive systematic review and meta-analysis to explore the bidirectional association between impaired lung function and hypertension therefore, this study provides new insights that may influence the care for patients with these two conditions. The large aggregated sample size increases the power and accuracy of pooled effect estimates, while the inclusion of studies with the same standard measurements of ILF and hypertension minimizes variability effects. Furthermore, the inclusion of the studies from multiple continents enhances the generalizability of the findings. A comprehensive search was done using explicit search strategies covering multiple databases without applying any language restrictions, which strengthened the completeness of the review and minimized the risk of selection or cultural bias, thereby not affecting the generalizability of the results. The review protocol was registered in PROSPERO, which enhanced transparency and methodological rigor. Another strength of this review is the effort taken to contact authors to request unpublished literature as to reduce publication bias. Sampling bias was reduced through independent and blind screening of titles and abstracts by two researchers. In addition, this meta-analysis included robust sensitivity and subgroup analyses to evaluate the findings of association between hypertension and impaired lung function and vice versa. Meta-regression was also attempted to explore potential sources of heterogeneity across the studies. The quality of the included studies was carefully assessed using NOS and JBI quality assessment tools. Hand-searching of references was even conducted to reduce the likelihood of missing relevant data. The implementation of trim-and-fill method further strengthened the assessment of potential publication bias. The inclusion of cohort study data allowed exploration of temporal relationships and potential causality when impaired lung function was the exposure.

However, the potential limitations of this meta-analysis should be considered. All of the included studies were observational in design, and the majority were cross-sectional, limiting causal inference. Although inclusion of a single cohort study provided evidence of a temporal relationship, residual and unmeasured confounding cannot be excluded and observed associations should not be interpreted as definitive evidence of causality. Important comorbidities such as smoking burden, obesity, underlying respiratory conditions and metabolic syndrome as well as important confounders such as level of physical activity, occupational or environmental exposures, and the effect of medication were not consistently reported across studies. As a result, comprehensive adjustment for these factors was not possible, and residual confounding cannot be excluded, limiting their inclusion in subgroup or meta regression analyses. Despite efforts to minimize heterogeneity through sensitivity, subgroup analyses, and meta-regression significant heterogeneity persisted across some studies. Lack of standardization of spirometry equipment and calibration across studies may have introduced variability in lung function measurements. Details regarding device type and calibration procedure were not consistently reported. Therefore, that limited our ability to assess this source of bias quantitatively. In addition, residual heterogeneity may reflect broader clinical and methodological diversity across studies, including differences in the definition of impaired lung function based on fixed ratio or lower limit of normal criteria, variability in spirometry reference equations, and differences in

population characteristics such as age distribution, comorbidity profiles, and geographic settings. Furthermore, residual heterogeneity may have arisen from differences in hypertension measurement protocols across studies, including variations in measurement devices, the timing and setting of blood pressure assessment, the number of readings obtained, and whether hypertension status was determined through direct measurement, medication use, or self-reported physician diagnosis. These methodological variations may have contributed to heterogeneity and could have influenced the pooled effect estimates. Therefore, pooled estimate should be interpreted as an average association across heterogeneous populations rather than a precise universal effect size. Despite efforts being made to include unpublished studies by contacting authors via email, relevant raw data could not be obtained. Consequently, studies with null or negative findings may be underrepresented, potentially leading to publication bias and an overestimation of the pooled effect size. This limitation cannot be fully excluded. Our study focused on odds ratios (ORs), excluding studies with other indices such as relative risk or hazard ratios, and the limited adjustment for confounders across studies hindered comprehensive exploration of heterogeneity. While the quality of included studies varied and publication bias was detected, these factors did not significantly affect the validity of the findings, though caution is warranted in interpreting the results.

## Conclusions

In conclusion, this systematic review and meta-analysis provides the first comprehensive evidence of a significant positive association between impaired lung function and hypertension in both directions. The available limited prospective evidence suggests impaired lung function contributes to subsequent hypertension risk, while support for the opposite causal direction is lacking. However, the significant heterogeneity observed between studies warrants cautious interpretation of these findings. These findings pave the way for future research and highlight important considerations for clinical and public health interventions. These findings further suggest that lung function impairment, particularly restrictive patterns, may represent an early marker of cardiovascular risk and warrant greater attention in cardiovascular screening. The observed bidirectional association also supports the possibility of shared inflammatory and metabolic mechanisms underlying both conditions. From clinical perspective, individuals with restrictive spirometry patterns may benefit from closer blood pressure monitoring, and all patients with hypertension may warrant periodic spirometry to assess lung function.

Moreover, the association between impaired lung function and increased risk of hypertension and the effect of hypertension on impaired lung function in this meta-analysis might be confounded by various factors. Therefore, future research should focus on identifying potential confounders specific to each condition, as well as those common to both, to ensure a more accurate interpretation of these findings. Large-scale, population based epidemiological studies are recommended to assess the impact of lung function impairment on hypertension and vice versa.

## Supporting information

**S1 File. PROSPERO protocol.**
(PDF)

**S2 File. Completed PRISMA 2020 checklist.**
(DOCX)

**S3 File. Completed PRISMA 2020 abstract checklist.**
(DOCX)

**S4 File. Newcastle – Ottawa quality assessment scale for cohort studies.**
(DOCX)

**S5 File. Newcastle – Ottawa quality assessment scale for case-control studies.**
(DOCX)

**S6 File. JBI critical appraisal checklist for analytical cross-sectional studies.**
(DOCX)

**S7 File. The csv file containing all screened records.**
(CSV)

**S1 Table. Results of the sensitivity analysis – ILF (exposure) and HT (outcome) – unadjusted analysis.**
(DOCX)

**S2 Table. Results of the sensitivity analysis – ILF (exposure) and HT (outcome) – adjusted analysis.**
(DOCX)

**S3 Table. Results of the sensitivity analysis – HT (exposure) and ILF (outcome) – unadjusted analysis.**
(DOCX)

**S4 Table. Summary of subgroup analyses for the adjusted ORs (exposure: impaired lung function; outcome: hypertension) to explore the source of heterogeneity.**
(DOCX)

**S5 Table. Summary of subgroup analyses for the adjusted ORs (exposure: hypertension; outcome: impaired lung function) to explore the source of heterogeneity.**
(DOCX)

## Acknowledgments

We extend our gratitude to all the authors who generously shared their data. We would also like to acknowledge K.T.A. Sandeeshwara Kasturiratna from Singapore Management University for her valuable contribution to the literature search.

## Author contributions

**Conceptualization:** Dilakshi Lekamge, Anuradhani Kasturiratne.

**Data curation:** Dilakshi Lekamge, Anuradhani Kasturiratne, Malay Kanti Mridha.

**Formal analysis:** Dilakshi Lekamge.

**Investigation:** Anuradhani Kasturiratne, Malay Kanti Mridha, John Chambers.

**Methodology:** Dilakshi Lekamge.

**Resources:** Dilakshi Lekamge, Anuradhani Kasturiratne.

**Software:** Dilakshi Lekamge.

**Supervision:** Anuradhani Kasturiratne, Malay Kanti Mridha, John Chambers.

**Validation:** Dilakshi Lekamge, Anuradhani Kasturiratne.

**Visualization:** Dilakshi Lekamge.

**Writing – original draft:** Dilakshi Lekamge.

**Writing – review & editing:** Anuradhani Kasturiratne, John Chambers.

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
