## [Decision Letter · Decision Letter 0]

30 Dec 2025

Dear Dr. Lekamge,

Thank you for submitting your manuscript to PLOS ONE. After careful consideration, we feel that it has merit but does not fully meet PLOS ONE’s publication criteria as it currently stands. Therefore, we invite you to submit a revised version of the manuscript that addresses the points raised during the review process.

We look forward to receiving your revised manuscript.

Kind regards,

George Kuryan

Academic Editor

PLOS One

Journal Requirements:

Reviewers' comments:

Reviewer's Responses to Questions

**Comments to the Author**

1. Is the manuscript technically sound, and do the data support the conclusions?

Reviewer #1: Yes

2. Has the statistical analysis been performed appropriately and rigorously?

Reviewer #1: I Don't Know

3. Have the authors made all data underlying the findings in their manuscript fully available?

Reviewer #1: Yes

4. Is the manuscript presented in an intelligible fashion and written in standard English?

Reviewer #1: Yes

Reviewer #1: The meta Analysis and review of literature is good and highlights the efforts taken by the authors to show the bidirectional association between impaired lung functions and hypertension. However, tgere is too much of reliance on heterogeneous observational studies and limits causal inference.There would have been lot of differences in comorbidities like smoking burden, obesity, underlying respiratory conditions, metabolic syndrome which may not have been accounted for.The authors are also not clear in their definition about impaired lung functions. The spirometers were not standardised and this may affect pooled estimates.Important confounders like level of physical activity, environmental or occupational exposure, effect of medications were not taken into account.Despite trying to retrieve a lot of raw data, null or negative studies may be underestimated.

.

Reviewer #1: No

---

## [Author Response · Author response to Decision Letter 1]

15 Jan 2026

Reviewer comment 1: The meta-analysis and review of literature is good and highlights the efforts taken by the authors to show the bidirectional association between impaired lung functions and hypertension.

Response for comment 1: We sincerely thank the reviewer for the positive feedback.

Reviewer comment 2: There is too much of reliance on heterogeneous observational studies and limits causal inference.

Response for comment 2: We fully agree with the reviewer that the observational nature of the included studies limits causal inference as our meta-analysis is primarily based on observational studies. Although the inclusion of a single cohort study allowed some assessment of temporal relationship, residual confounding cannot be excluded, and definitive causality can not be established. We have clarified this point in the abstract and the discussion of the revised manuscript and emphasized that our findings indicate associations rather than causal relationships.

The conclusion section of the abstract has been updated with the following text: “This study concludes that there is a significant positive relationship between impaired lung function and hypertension, with temporal evidence suggesting that impaired lung function may precede hypertension, while the reverse causal direction lacks such evidence.”

The discussion section was updated with the following text: “All of the included studies were observational in design; therefore, causal inference is limited. Although inclusion of a single cohort study provided evidence of a temporal relationship, residual and unmeasured confounding cannot be excluded and observed associations should not be interpreted as definitive evidence of causality.”

Reviewer comment 3: There would have been lot of differences in comorbidities like smoking burden, obesity, underlying respiratory conditions, metabolic syndrome which may not have been accounted for.

Response for comment 3: We agree that these comorbidities may influence both lung function and hypertension. Unfortunately, these variables were not uniformly reported across studies, limiting our ability to perform adjusted pooled analyses. However, wherever available, we extracted multivariable-adjusted estimates that account for major confounders. We have now clearly acknowledged in the discussion that residual confounding from such comorbidities remains possible.

The discussion section was revised with the following text: “Important comorbidities such as smoking burden, obesity, underlying respiratory conditions and metabolic syndrome were not uniformly reported across studies. As a result, comprehensive adjustment for these factors was not possible, and residual confounding cannot be excluded.”

Reviewer comment 4: The authors are also not clear in their definition about impaired lung functions.

Response for comment 4: We thank the reviewer for this comment. We agree that our original description of impaired lung function lacked clarity and we have now revised it under the selection criteria of the methods section as follows: “We only accepted the articles where, impaired lung function was defined according to the criteria used in the individual studies, provided that impairment was based on FEV1 (forced expiratory volume in 1 second), FVC (forced vital capacity) and FEV1/FVC spirometric measurements. Studies were included if they reported impaired lung function as obstructive, restrictive, or mixed pattern, or if impairment was defined using accepted thresholds such as FEV1/FVC < 0.70 or values below the lower limit of normal (LLN, <5th percentile) of the FEV1/FVC ratio calculated from the reference equations.”

Reviewer comment 5: The spirometers were not standardised, and this may affect pooled estimates.

Response for comment 5: We agree that lack of standardization of spirometry equipment and calibration across studies may have introduced variability in lung function measurements. Details regarding device type and calibration procedures were not consistently reported. Therefore, that limited our ability to assess this source of bias quantitatively. These methodological variations may have contributed to heterogeneity and could have influenced the pooled effect estimates. We have now added this clarification to the limitations paragraph of the strengths and limitations under the discussion.

Reviewer comment 6: Important confounders like level of physical activity, environmental or occupational exposure, effect of medications were not taken into account.

Response for comment 6: We agree with this important point. These variables were inconsistently reported across studies and therefore could not be comprehensively included in subgroup or meta regression analyses. We have now added this under limitations acknowledging the potential influence of unmeasured or residual confounding related to physical activity, occupational or environmental exposures, and effect of medication.

Reviewer comment 7: Despite trying to retrieve a lot of raw data, null or negative studies may be underestimated.

Response for comment 7: We agree with the reviewer that studies with null or negative findings may be under-represented in the available literature, despite our efforts to obtain unpublished data from authors. This under representation may contribute to publication bias and could potentially lead to an overestimation of the pooled effect size. To minimize this risk, we conducted comprehensive database searches, reviewed reference lists, and attempted to retrieve unpublished data where possible. We have now strengthened the limitations section to reflect that the under-representation of null or negative studies cannot be fully excluded.

---

## [Decision Letter · Decision Letter 1]

16 Feb 2026

Dear Dr. Lekamge,

Thank you for submitting your manuscript to PLOS ONE. After careful consideration, we feel that it has merit but does not fully meet PLOS ONE’s publication criteria as it currently stands. Therefore, we invite you to submit a revised version of the manuscript that addresses the points raised during the review process.

We look forward to receiving your revised manuscript.

Kind regards,

George Kuryan

Academic Editor

PLOS One

Journal Requirements:

Reviewers' comments:

Reviewer's Responses to Questions

**Comments to the Author**

Reviewer #1: All comments have been addressed

Reviewer #2: (No Response)

2. Is the manuscript technically sound, and do the data support the conclusions?

Reviewer #1: Yes

Reviewer #2: Partly

3. Has the statistical analysis been performed appropriately and rigorously?

Reviewer #1: Yes

Reviewer #2: No

4. Have the authors made all data underlying the findings in their manuscript fully available?

Reviewer #1: Yes

Reviewer #2: Yes

5. Is the manuscript presented in an intelligible fashion and written in standard English?

Reviewer #1: Yes

Reviewer #2: Yes

Reviewer #1: All comments have been addressed to satisfaction and the manuscript may be accepted if found satisfactory by other members of the reviewing committee and the editorial board.

Reviewer #2: This manuscript presents a comprehensive and timely systematic review and meta-analysis examining the bidirectional association between impaired lung function (ILF) and hypertension (HTN). The work is ambitious in scope, methodologically structured, and clinically relevant. The inclusion of both directions of association and the attempt to explore heterogeneity through subgroup and meta-regression analyses are notable strengths.

The manuscript makes a meaningful contribution to cardiovascular–respiratory epidemiology. However, several methodological clarifications, interpretative refinements, and structural improvements are required before publication.

MAJOR COMMENTS

1. Conceptual Framing of “Bidirectional” Association

Concern

The manuscript repeatedly refers to a “bidirectional relationship,” yet:

The majority of included studies are cross-sectional.

Only one cohort study examined ILF → HTN temporality.

There is no longitudinal evidence supporting HTN → ILF direction.

Thus, “bidirectional association” may overstate the strength of causal inference.

Recommendation

Rephrase to:

“Evidence supports a significant association in both directions, with stronger temporal evidence for impaired lung function preceding hypertension.”

In the Discussion and Conclusion, clearly distinguish:

Association (supported)

Causality (not established except limited evidence for ILF → HTN)

2. Clinical Interpretation of Effect Sizes

Key Findings:

ILF → HTN (adjusted OR): 1.40

HTN → ILF (adjusted OR): 1.94

Concern

The manuscript suggests hypertension has a “more substantial association” because OR is numerically higher. However:

Different sample sizes

Different heterogeneity profiles

Stronger publication bias in HTN → ILF crude models

Only 6 adjusted studies in HTN → ILF analysis

Direct comparison of magnitude may be misleading.

Recommendation

Add clarification:

“Although adjusted effect sizes were numerically larger when hypertension was the exposure, differences in study design, number of included studies, and publication bias limit direct comparison of magnitude between directions.”

3. High Heterogeneity (I² > 80%)

Several analyses show substantial heterogeneity:

ILF → HTN (crude): I² = 84%

HTN → ILF (crude): I² = 84.7%

Concern

Despite subgroup and meta-regression analyses, residual heterogeneity remains high.

Recommendations

Provide clearer interpretation:

Clinical heterogeneity (definitions of ILF)

Population differences

Variability in spirometry reference equations

Differences in hypertension measurement protocols

Explicitly state:

“The pooled estimate should be interpreted as an average association across heterogeneous populations rather than a precise universal effect size.”

4. Restrictive Lung Impairment Signal

You consistently report that restrictive impairment shows stronger association with HTN.

This is clinically important and deserves deeper discussion.

Suggested Expansion

Discuss potential mechanisms:

Obesity-related restrictive physiology

Metabolic syndrome clustering

Low-grade systemic inflammation

Diaphragmatic mechanics

Vascular stiffness and reduced lung compliance

Currently, this finding is mentioned but not mechanistically elaborated enough.

This could become one of the manuscript’s key contributions.

5. Publication Bias Interpretation

For HTN → ILF (crude):

Egger’s test significant

Trim-and-fill changes significance

Yet you conclude:

“Adjustment does not change overall conclusion.”

This is inaccurate because:

After trim-and-fill, p = 0.280 (non-significant)

Recommendation

Revise to:

“After adjusting for potential publication bias, the crude association between hypertension and impaired lung function lost statistical significance, suggesting that small-study effects may have inflated the initial estimate.”

This is important for credibility.

6. Meta-Regression Interpretation

Your multivariable meta-regression:

Explained 32.56% heterogeneity

No moderators statistically significant

Clarification Needed

The manuscript suggests sex and education reduce effect size in univariate models but then are not significant in multivariable analysis.

Explain:

Collinearity

Reduced power

Limited study-level variability

Otherwise, interpretation appears inconsistent.

MODERATE COMMENTS

7. Overstatement of Causality

Statements such as:

“This evidence suggests a potential causal pathway…”

Should be softened to:

“This pattern is consistent with a potential causal pathway…”

Observational data cannot establish causation definitively.

8. Definition Variability

Hypertension definition:

SBP ≥ 140 / DBP ≥ 90

Self-report

Medication use

ILF definition:

Fixed ratio vs LLN

GOLD vs study-specific cutoffs

This heterogeneity likely affects pooled results. A dedicated paragraph discussing fixed ratio vs LLN bias would strengthen the methodological rigor.

9. Terminology Precision

Minor but important:

Use “odds ratio” not “odd ratio”

Use consistent abbreviation for impaired lung function (ILF)

Avoid redundancy in abstract (“approximately a significant one-fold higher risk” → simply “significantly higher risk”)

MINOR COMMENTS

Remove repetition in Discussion about confounders.

Consider reducing extensive funnel plot narration; condense.

Add a summary table comparing ILF → HTN vs HTN → ILF for clarity.

Correct grammar inconsistencies throughout (copy-edit recommended).

STRENGTHS OF THE STUDY

First comprehensive bidirectional meta-analysis.

Large aggregated sample size.

Inclusion of multiple continents.

Robust sensitivity analyses.

Careful quality assessment (NOS & JBI).

Attempted meta-regression.

Registration on PROSPERO.

Trim-and-fill implementation.

CLINICAL & PUBLIC HEALTH SIGNIFICANCE

This study suggests:

Lung function impairment may be an early cardiovascular risk marker.

Restrictive physiology deserves attention in cardiovascular screening.

Shared inflammatory and metabolic pathways may underlie both conditions.

For clinicians:

Patients with restrictive spirometry patterns may warrant BP monitoring.

Hypertensive patients may benefit from periodic spirometry.

.

Reviewer #1: No

Reviewer #2: No

---

## [Author Response · Author response to Decision Letter 2]

7 Mar 2026

Reviewer 1 overall comment: All comments have been addressed to satisfaction, and the manuscript may be accepted if found satisfactory by other members of the reviewing committee and the editorial board.

Response for Reviewer 1 overall comment: We sincerely thank the reviewer for the positive feedback on our manuscript and for confirming that all previous comments have been satisfactorily addressed.

Reviewer 2 overall comment: This manuscript presents a comprehensive and timely systematic review and meta-analysis examining the bidirectional association between impaired lung function (ILF) and hypertension (HTN). The work is ambitious in scope, methodologically structured, and clinically relevant. The inclusion of both directions of association and the attempt to explore heterogeneity through subgroup and meta-regression analyses are notable strengths. The manuscript makes a meaningful contribution to cardiovascular–respiratory epidemiology. However, several methodological clarifications, interpretative refinements, and structural improvements are required before publication.

Response for Reviewer 2 comment: We thank the reviewer for the thoughtful and constructive feedback. We have addressed each comment in detail below.

Reviewer 2 comment 1: Conceptual Framing of “Bidirectional” Association

The manuscript repeatedly refers to a “bidirectional relationship,” yet: The majority of included studies are cross-sectional. Only one cohort study examined ILF → HTN temporality. There is no longitudinal evidence supporting HTN → ILF direction. Thus, “bidirectional association” may overstate the strength of causal inference.

Recommendation: Rephrase to: “Evidence supports a significant association in both directions, with stronger temporal evidence for impaired lung function preceding hypertension.” In the Discussion and Conclusion, clearly distinguish: Association (supported), Causality (not established except limited evidence for ILF → HTN)

Response for Reviewer 2 comment 1: We agree with the reviewer that the term “bidirectional association” may overstate what is supported by the available evidence. We have clarified this point in the abstract, discussion, and the conclusion of the revised manuscript and emphasized that our findings indicate associations rather than causal relationships.

The conclusion section of the abstract has been revised with the following text: “This study concludes that there is a significant positive association between impaired lung function and hypertension in both directions, with stronger temporal evidence suggesting that impaired lung function may precede hypertension, while the reverse causal direction lacks such evidence.”

The discussion section was updated with the following text: “Since cross-sectional studies do not support temporal associations, the findings should be interpreted cautiously. Limited longitudinal evidence from a single cohort study suggests that impaired lung function may precede hypertension; however, comparable longitudinal evidence is lacking to support temporality in the reverse direction. Therefore, these results support significant associations in both directions but do not establish causality except limited evidence when impaired lung function is considered the exposure.”

We have revised the limitations section to clarify that causal inference is limited due to the predominance of cross-sectional studies and to emphasize that longitudinal evidence supporting temporality is limited to impaired lung function preceding hypertension. The limitation section under the discussion section has been revised with the following text: “All of the included studies were observational in design, and the majority were cross-sectional, limiting causal inference.”

The conclusion section of paper has been revised with the following text: “In conclusion, this systematic review and meta-analysis provides the first comprehensive evidence of a significant positive association between impaired lung function and hypertension in both directions. The available limited prospective evidence suggests impaired lung function contributes to subsequent hypertension risk, while support for the opposite causal direction is lacking.”

Reviewer 2 comment 2: Clinical Interpretation of Effect Sizes

Key Findings: ILF → HTN (adjusted OR): 1.40, HTN → ILF (adjusted OR): 1.94

Concern: The manuscript suggests hypertension has a “more substantial association” because OR is numerically higher. However: Different sample sizes, Different heterogeneity profiles, Stronger publication bias in HTN → ILF crude models, Only 6 adjusted studies in HTN → ILF analysis, Direct comparison of magnitude may be misleading.

Recommendation: Add clarification: “Although adjusted effect sizes were numerically larger when hypertension was the exposure, differences in study design, number of included studies, and publication bias limit direct comparison of magnitude between directions.”

Response for Reviewer 2 comment 2: We totally agree with the above concern, and we have revised the Discussion section of the paper to clarify that although adjusted effect sizes were numerically larger when hypertension was the exposure, differences in study characteristics and potential publication bias limit direct comparison of magnitude between directions. The Discussion has been revised with the following text: “Although adjusted effect sizes were numerically larger when hypertension was considered as the exposure, differences in study design, number of included studies, and publication bias limit direct comparison of magnitude between the two directions.”

Reviewer 2 comment 3: High Heterogeneity (I² > 80%)

Several analyses show substantial heterogeneity: ILF → HTN (crude): I² = 84%

HTN → ILF (crude): I² = 84.7%

Concern: Despite subgroup and meta-regression analyses, residual heterogeneity remains high.

Recommendations: Provide clearer interpretation: Clinical heterogeneity (definitions of ILF), Population differences, Variability in spirometry reference equations, Differences in hypertension measurement protocols

Explicitly state: “The pooled estimate should be interpreted as an average association across heterogeneous populations rather than a precise universal effect size.”

Response for Reviewer 2 comment 3: We thank the reviewer for this valuable comment. In response, we have expanded the Discussion to provide a clearer interpretation of the residual heterogeneity observed in our analyses. Specifically, we have elaborated on potential sources of clinical, methodological, and population heterogeneity, including differences in definitions of impaired lung function, variability in spirometry reference equations, differences in population characteristics, and variations in hypertension measurement protocols.

The following text has been added to the Discussion section of the manuscript: “In addition, residual heterogeneity may reflect broader clinical and methodological diversity across studies, including differences in the definition of impaired lung function based on fixed ratio or lower limit of normal criteria, variability in spirometry reference equations, and differences in population characteristics such as age distribution, comorbidity profiles, and geographic settings. Furthermore, residual heterogeneity may have arisen from differences in hypertension measurement protocols across studies, including variations in measurement devices, the timing and setting of blood pressure assessment, the number of readings obtained, and whether hypertension status was determined through direct measurement, medication use, or self-reported physician diagnosis. These methodological variations may have contributed to heterogeneity and could have influenced the pooled effect estimates. Therefore, pooled estimate should be interpreted as an average association across heterogeneous populations rather than a precise universal effect size.”

Reviewer 2 comment 4: Restrictive Lung Impairment Signal

You consistently report that restrictive impairment shows stronger association with HTN.

This is clinically important and deserves deeper discussion.

Suggested Expansion: Discuss potential mechanisms:

Obesity-related restrictive physiology

Metabolic syndrome clustering

Low-grade systemic inflammation

Diaphragmatic mechanics

Vascular stiffness and reduced lung compliance

Currently, this finding is mentioned but not mechanistically elaborated enough.

This could become one of the manuscript’s key contributions.

Response for Reviewer 2 comment 4: We thank the reviewer for this valuable suggestion. We agree that the stronger association observed between restrictive lung impairment and hypertension is clinically important and warrants further mechanistic discussion. Accordingly, we have expanded the Discussion section to provide a more detailed explanation of potential biological pathways underlying this relationship. We have specifically elaborated on obesity related restrictive respiratory physiology, including reductions in lung volumes and mechanical limitations of the thoraco-abdominal system, the role of metabolic syndrome clustering, the contribution of low-grade systemic inflammation driven by adipose tissue dysfunction, the impact of altered diaphragmatic mechanics and the effects of vascular stiffness and reduced lung compliance.

These additions provide biological plausibility for our findings and highlight the potential role of restrictive lung impairment as a clinical marker of systemic cardiometabolic dysfunction and provide a descriptive explanation on why restrictive impairment may demonstrate a more consistent and robust association with hypertension compared to obstructive impairment.

The revised text has been added to the Discussion section of the revised manuscript with track changes (Page 40 and 41, Lines 850 – 889). We have also highlighted this observation as one of the key contributions of the present study.

Reviewer 2 comment 5: Publication Bias Interpretation

For HTN → ILF (crude): Egger’s test significant, Trim-and-fill changes significance, Yet you conclude: “Adjustment does not change overall conclusion.” This is inaccurate because: After trim-and-fill, p = 0.280 (non-significant)

Recommendation: Revise to: “After adjusting for potential publication bias, the crude association between hypertension and impaired lung function lost statistical significance, suggesting that small-study effects may have inflated the initial estimate.” This is important for credibility.

Response for Reviewer 2 comment 5: We thank the reviewer for this important observation. We agree that our original wording did not accurately reflect the impact of potential publication bias on the crude association between hypertension and impaired lung function. Accordingly, we have now revised the relevant section of the manuscript to clarify that the Egger’s test indicated possible small-study effects and after applying the trim-and-fill method, the pooled crude association was no longer statistically significant.

The revised section is as follows: “After adjusting for potential publication bias, the crude association between hypertension and impaired lung function lost statistical significance, suggesting that small-study effects may have inflated the initial estimate.”

Reviewer 2 comment 6: Meta-Regression Interpretation

Your multivariable meta-regression:

Explained 32.56% heterogeneity

No moderators statistically significant

Clarification Needed

The manuscript suggests sex and education reduce effect size in univariate models but then are not significant in multivariable analysis. Explain: Collinearity, Reduced power, Limited study-level variability. Otherwise, interpretation appears inconsistent.

Response for Reviewer 2 comment 6: We agree with the reviewer that clarification is needed regarding the interpretation of univariate and multivariable meta-regression findings. We have revised the Discussion section to clarify that although sex and education were significant moderators in the univariate analyses, they were not statistically significant in the multivariable model. It has now been clearly stated that this inconsistency can be explained by collinearity among study-level characteristics, reduced statistical power due to limited number of studies, and limited variability of moderators across studies.

The following revised section has been added to the Discussion section: “Although sex and education were significantly associated with effect size in univariate meta-regression, these associations were no longer statistically significant in the multivariable model despite explaining a substantial proportion of heterogeneity. The loss of statistical significance in the multivariable meta-regression likely reflects collinearity among study-level characteristics such as sex and education, reduced statistical power due to the modest number of included studies relative to the number of moderators examined, and limited variability of several moderators across studies. These factors may have reduced the ability of the model to detect independent moderator effects despite their apparent influence in univariate analyses.”

Moderate Comments

Reviewer 2 comment 7: Overstatement of Causality

Statements such as: “This evidence suggests a potential causal pathway…”

Should be softened to: “This pattern is consistent with a potential causal pathway…”

Observational data cannot establish causation definitively.

Response for Reviewer 2 comment 7: We thank the reviewer for this valuable comment. We agree that causal interpretations should be made cautiously, given that the included studies were observational in design. Accordingly, we have revised the Discussion section to remove statements implying causality and clarified that the findings indicate significant associations rather than definitive causal relationships.

Reviewer 2 comment 8: Definition Variability

Hypertension definition: SBP ≥ 140 / DBP ≥ 90, Self-report, Medication use

ILF definition: Fixed ratio vs LLN, GOLD vs study-specific cutoffs

This heterogeneity likely affects pooled results. A dedicated paragraph discussing fixed ratio vs LLN bias would strengthen the methodological rigor.

Response for Reviewer 2 comment 8: We thank the reviewer for this important and constructive comment. As noted, we have previously addressed the broader issue of clinical and methodological heterogeneity in response to Reviewer 2 comment 3. In response to this additional suggestion, we have now further strengthened the Discussion section by adding a dedicated paragraph specifically addressing the potential bias associated with the use of fixed ratio versus lower limit of normal (LLN) criteria for defining impaired lung function. In addition, we have included further clarification on definition variability across studies, including differences in hypertension and variability in the classification of impaired lung function using GOLD classification versus study-specific spirometric cutoff values.

The revised text is as follows: “Although, most studies defined hypertension using the widely accepted clinical threshold of systolic blood pressure ≥ 140 mmHg and/or diastolic blood pressure ≥ 90 mmHg, some incorporated antihypertensive medication use or self-reported physician diagnosis to identify treated or previously diagnosed hypertensives, which may introduce differences in definition. Similarly, an important source of methodological variability across studies relates to differences in the definition of impaired lung function. While some studies defined impairment using the fixed ratio threshold (FEV1/FVC < 0.70), others applied the lower limit of normal (LLN, <5th percentile) of the FEV1/FVC ratio based on age, sex, and population specific reference equations. These differing definitions can lead to systematic differences in case classification across studies. The fixed ratio approach does not account for the physiological decline in lung function with aging and may therefore overestimate impairment in older populations while underestimating it in younger individuals, resulting in potential age-related misclassification bias. In contrast, LLN provides individualized thresholds and re

---

## [Decision Letter · Decision Letter 2]

22 Mar 2026

Association between hypertension and impaired lung function among adults: A systematic review and meta-analysis

PONE-D-25-57756R2

Dear Dr.Lekamge

We’re pleased to inform you that your manuscript has been judged scientifically suitable for publication and will be formally accepted for publication once it meets all outstanding technical requirements.

Kind regards,

Kuryan George

Academic Editor

PLOS One

Additional Editor Comments (optional):

Reviewers' comments:

Reviewer's Responses to Questions

**Comments to the Author**

Reviewer #1: All comments have been addressed

Reviewer #2: All comments have been addressed

2. Is the manuscript technically sound, and do the data support the conclusions?

Reviewer #1: Yes

Reviewer #2: (No Response)

3. Has the statistical analysis been performed appropriately and rigorously?

Reviewer #1: Yes

Reviewer #2: (No Response)

4. Have the authors made all data underlying the findings in their manuscript fully available?

Reviewer #1: Yes

Reviewer #2: (No Response)

5. Is the manuscript presented in an intelligible fashion and written in standard English?

Reviewer #1: Yes

Reviewer #2: (No Response)

Reviewer #1: All comments have been addressed to satisfaction and may be accepted if found suitable by the editorial board.

Reviewer #2: (No Response)

.

Reviewer #1: No

Reviewer #2: No

---

## [Editor Report · Acceptance letter]

PONE-D-25-57756R2

PLOS One

Dear Dr. Lekamge,

I'm pleased to inform you that your manuscript has been deemed suitable for publication in PLOS One. Congratulations! Your manuscript is now being handed over to our production team.

Kind regards,

on behalf of

Professor George Kuryan

Academic Editor

PLOS One